

# The relative importance of macrophysical and cloud albedo changes for aerosol induced radiative effects in stratocumulus

Daniel P. Grosvenor[1], Paul R. Field[1,2], Adrian A. Hill[2], and Benjamin J. Shipway[2]

[1]School of Earth and Environment, University of Leeds, Leeds, LS2 9JT, UK
[2]Met Office, Exeter, UK

*Correspondence to:* D. P. Grosvenor
(daniel.p.grosvenor@gmail.com)

**Abstract.** Aerosol-cloud interactions are explored using 1 km resolution simulations of SE Pacific stratocumulus clouds that include realistic meteorology along with newly implemented cloud microphysics and sub-grid cloud schemes. The model was critically assessed against observations of Liquid Water Path (LWP), broadband fluxes, cloud fraction ($f_c$), droplet number concentrations ($N_d$) and radar reflectivities.

Aerosol loading sensitivity tests showed that at low aerosol loadings, changes to aerosol affected shortwave fluxes equally through changes to cloud macrophysical charateristics (LWP, $f_c$) and cloud albedo changes due solely to $N_d$ changes. However, at high aerosol loadings, only the $N_d$ albedo change was important. Evidence was also provided to show that a treatment of sub-grid clouds is as important as order of magnitude changes in aerosol loading for the accurate simulation of stratocumulus at this grid resolution.

Overall, the control model demonstrated a credible ability to reproduce observations suggesting that many of the important physical processes for accurately simulating these clouds are represented within the model and giving some confidence in the predictions of the model concerning stratocumulus and the impact of aerosol. For example, the control run was able to reproduce the shape and magnitude of the observed diurnal cycle of domain mean LWP to within $\sim$10 $gm^{-2}$ for the nighttime, but with an overestimate for the daytime of up to 30 $gm^{-2}$. The latter was attributed to the uniform aerosol fields imposed on the model, which meant that the model failed to include the low $N_d$ mode that was observed further offshore, preventing the LWP removal through precipitation that likely occurred in reality. The boundary layer was too low by around 260 m, which was attributed to the driving global model analysis. The shapes and sizes of the observed bands of clouds and open–cell–like regions of low areal cloud cover were qualitatively captured. The daytime $f_c$ frequency distribution was reproduced to within $f_c$=0.04 for $f_c>\sim$0.7 as was the domain mean nighttime $f_c$ (at a single time) to within $f_c$=0.02. Frequency distributions of shortwave top-of-the-atmosphere (TOA) fluxes from satellite were well represented by the model with only a slight underestimate of the mean by 15 %; this was attributed to near–shore aerosol concentrations that were too low for the particular times of the satellite overpasses. TOA longwave flux distributions were close to those from satellite with agreement of the mean value to within 0.4 %. From comparisons of $N_d$ distributions to those from satellite it was found that the $N_d$ mode from the model agreed with the higher of the two observed modes to within $\sim$15 %.


## 1 Introduction

In this paper we describe 1 km horizontal grid-spacing simulations of marine stratocumulus clouds nested within a global operational analysis framework that provides realistic meteorological initial conditions and lateral boundary forcing. A grid-spacing of this order bridges the gap between LES (Large Eddy Simulation) and global model resolution, allowing larger domains than

possible with LES, but the direct representation of more detailed processes than is possible with global models. We perform the first tests for stratocumulus of a newly implemented microphysics package that includes a detailed representation of the effects of aerosol upon clouds and a diagnostic cloud scheme. We use this model to examine the response of the cloud field to varying aerosol concentrations.

Stratocumulus clouds are the dominant cloud type in terms of area, covering over one fifth of the Earth's surface in the

annual mean (Wood, 2012). They exert a strong net negative radiative effect that has a major impact on Earth's radiative balance (Hartmann et al., 1992) and only a small change in their properties would have a large radiative impact (e.g. Latham et al., 2008). The albedo and the spatial coverage of stratocumulus clouds are affected by both their macrophysical and microphysical properties with aerosol potentially playing a key role in modulating both of these aspects. If this is the case then the accurate representation of cloud aerosol interactions would be needed in order to make robust predictions about the response

of stratocumulus to climate change and anthropogenic aerosol changes. Furthermore, since uncertainties in the representation of stratocumulus have been identified as one of the major sources of uncertainty in climate model predictions (Bony, 2005; Soden and Vecchi, 2011), it follows that the treatment of aerosol will influence this uncertainty if the aerosol has a significant cloud impact.

Stratocumulus is also important for Numerical Weather Prediction (NWP) because it modulates the surface temperature

through its influence on downwelling shortwave and longwave radiation at the surface. Further its influence on visibility is a major consideration for aircraft operations. There is therefore a strong impact on both commercial and general public weather forecasts and applications.

For the climate system, the radiative impact of stratocumulus is strongly dependent on macrophysical properties such as cloud fraction or cloud Liquid Water Path (LWP), which are likely to be heavily influenced by the large scale circulation and

meteorological factors. However, microphysical processes can also influence the macrophysical cloud properties, as well as having important radiative impacts in their own right. If all else is equal, i.e. a fixed liquid water content (LWC), increasing the concentration of Cloud Condensation Nuclei (CCN) leads to smaller droplets that in turn produce more reflective clouds (Twomey, 1977). The reduction of droplet sizes is also associated with the suppression of precipitation. Since this removes the main sink for water in a cloud it was suggested that precipitation suppression via increases in aerosol would increase

LWP and cloud lifetime (e.g. Albrecht, 1989), an idea that has been backed up by LES modelling studies (Berner et al., 2013; Feingold et al., 2015; Ackerman et al., 2004, hereafter A04). However, A04 showed that this is only true for precipitating clouds; once precipitation had been suppressed, further aerosol increases led to cloud thinning (LWP decrease) via increases in entrainment. Mechanisms for this effect are discussed in Bretherton et al. (2007) and Hill et al. (2009). Observation studies





have also demonstrated a lack of LWP increase in marine stratocumulus at high aerosol concentrations (e.g. Ackerman et al., 2000; Platnick et al., 2000; Coakley and Walsh, 2002).

Changes in precipitation and LWP that result from changes in aerosol can also be accompanied by changes in cloud fraction (Stevens et al., 1998; Berner et al., 2013, hereafter B13). An example of this is the occurrence of Pockets of Open Cells

(POCs). POCs constitute regions of open cells with low cloud fraction in amongst high cloud fraction closed cell regions (Wood et al., 2011a). It has been suggested that the enhancement of precipitation by reduced aerosol concentrations can cause a transition between a state of closed and open cells within stratocumulus (Rosenfeld et al., 2006), which is then enhanced by a positive feedback mechanism that has been called the "runaway precipitation sink" (Feingold and Kreidenweis, 2002) whereby precipitation leads to a reduction in the available CCN. All else being equal, reducing CCN leads to larger drops that enhance

the formation of precipitation, promoting the removal of more CCN. High resolution idealised LES modelling supports this idea (B13) and shows that these processes occur at smaller spatial scales than can be captured explicitly by GCMs.

A compromise between LES and GCMs is a coarser resolution ($\sim$ 1km) regional model that can simulate larger domains for the same or less computational cost as an LES. Regional models have the advantage over LES in that they are driven by meteorological analyses that can capture the relevant large scale dynamic and thermodynamic structure, allowing results to be

more easily compared to real observations. Aerosol effects can also be considered relative to dynamical forcing or meteorology effects.

It is an open question whether km-scale grid spacings are adequate to simulate the important processes involved in marine stratocumulus. For example, Boutle and Abel (2012, hereafter BA12) showed that a mesoscale model with a 1 km gird spacing could capture closed cell stratocumulus well, but they did not look at open–cell behaviour. Results from WRF-CHEM at coarser

grid-spacings (9km, Yang et al. (2011); 12 km, Saide et al. (2012); 14 km, George et al. (2013)), where the representation of stratocumulus is reliant on boundary layer parameterizations, have also showed reasonable agreement with observations. Whilst the coarser resolution models may capture the general features of closed cell stratocumulus, the simulation of open cells is likely to be more difficult owing to the smaller size of the precipitating and updraft regions and the small scales over which aerosol-cloud interactions occur. It is unclear whether the combination of boundary layer parameterizations and microphysics

schemes used in the coarser models will encapsulate the correct response to aerosols.

In this paper we present results using a regional nested configuration of the Met Office Unified Model (UM). It is driven by realistic meteorology and includes a new microphysics scheme called CASIM (Cloud AeroSol Interaction Microphysics, see Section 2.1.2 for details) designed to simulate the processes important to aerosol–cloud interactions. Simulating a well-observed case allows the critical assessment of the model against a wide range of relevant observations. By demonstrating

that the model is capable of reproducing the observations we can argue that the model captures the important physics and will provide a reliable baseline for predicting the influence of aerosol on this stratocumulus cloud system.

Thus, we aim to address the following questions :-

1. Can a regional model produce a realistic representation of stratocumulus cloud when compared to a diverse range of observations?



2. How do the modelled clouds respond to aerosol?

3. What is the relative importance of macrophysical and cloud albedo changes for aerosol induced radiative effects?

4. What is the relative importance of the sub-grid cloud scheme?

## 2 Data and Methods

5 For this case study we simulate a near-coastal region of the SE Pacific (see Fig. 1) for the period 12-14th November, 2008, during which time mostly closed cell stratocumlus were observed. This period coincides with the VOCALS field campaign, which took place in this region and provided a variety of cloud, aerosol and meteorological measurements made from airborne, ship, radiosonde and buoy observational platforms (Wood et al., 2011b). A variety of satellite data is also available. Further details of the simulations and the observations used are now described.

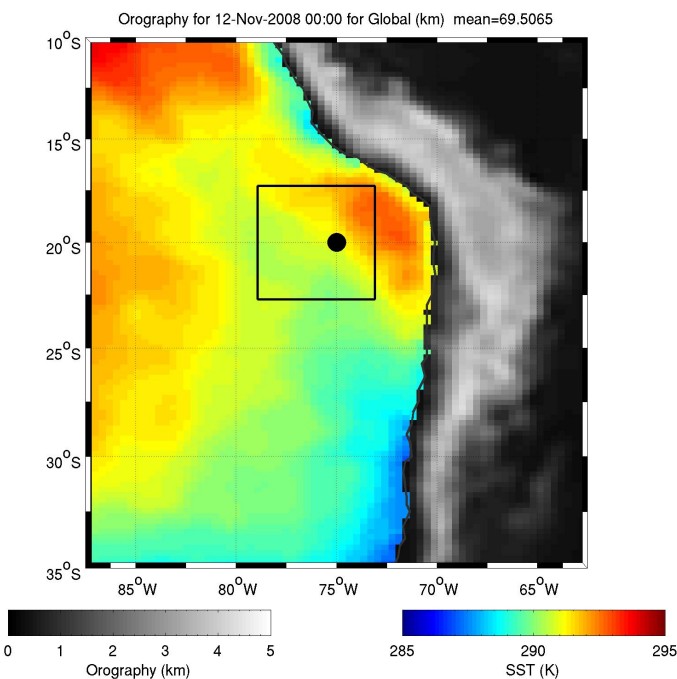

**Figure 1.** A map of the SE Pacific region with the 1 km model domain shown as a black box. The colours show the orography over land and the sea-surface temperature over the ocean, both at the resolution of the global model (N512; ∼39 km × 26 km resolution at the equator for dx×dy). The black dot shows the location of the RH Brown ship (20° S, 75° W).





## 2.1 Model details

In this study we use the NWP (Numerical Weather Prediction) configuration of the UK Met Office Unified Model (hereafter the UM). The global model used here is the GA6 configuration of the UM at N512 resolution (~39 km × 26 km resolution at the equator for dx×dy) with 70 vertical levels below 80 km that are quadratically spaced giving more levels near the surface. This

is used to drive a single 1 km resolution nest centred at 20° S, 76° W of size 600×600 km (Fig. 1). This places the domain near to the coast, but ensures that it covers only oceanic grid-points, which reduces the dynamical and computational complexity. The domain sits in the heart of the region where the VOCALS field campaign aircraft measurements took place (mostly in a transect along 20° S between the coast and 90° W) and means that the location of the Ronald H. Brown research vessel during the period (20° S, 75° W) is near the centre of the domain (see Fig. 1 for the location of the ship). The region consists of very

high and steep orography on the coast with the model predicting warm SSTs just offshore of the coast at the latitudes of the model domain, which reduce with distance offshore until just west of 80° W when they start to increase again.

The 1 km inner nest also employs 70 vertical levels, but with a lower domain top of 40 km and thus a higher vertical resolution. Table 1 shows that the vertical resolution near the top of the boundary layer for the inner nest (~1–1.5 km) is around 100–140 m. The 1 km nest uses a rotated pole coordinate system whose equator is situated at the centre of the domain.

A parametrized convection scheme is not required at high resolution since the model is likely to be convection permitting.

The global simulation uses the operational microphysics scheme based on Wilson and Ballard (1999), which is a single moment scheme in that it does not represent the number concentrations of hydrometeors. For the 1 km nest runs we primarily use the newly implemented double moment CASIM aerosol scheme that is described in Section 2.1.2.

### 2.1.1 A sub-grid cloud scheme

The recent previous studies of stratocumulus with the UM that employed high resolution nests e.g. BA12 used a sub-grid cloud scheme (Smith, 1990) that was linked to the Wilson and Ballard (1999) microphysics scheme. The sub-grid cloud scheme parameterizes the variability in relative humidity (RH) that occurs in reality within a grid-box, which may allow cloud to form even if the mean grid-box RH is below 100%. This can be important for stratocumulus since the presence of a some liquid cloud water generates longwave cooling at cloud top, which creates instability within the boundary layer. This drives turbulent

overturning that can in turn create more cloud (i.e. a positive feedback).

When CASIM was implemented into the UM, it was done so with no sub-grid cloud scheme. In this configuration there was a large under prediction in the amount of stratocumulus (see Section 3.2.2). Therefore, work was undertaken to implement and adapt the Smith (1990) approach to allow it to work with a multi-moment bulk scheme such as CASIM. Details of this implementation are provided in Appendix A.



### 2.1.2 The CASIM microphysics scheme

CASIM (Cloud AeroSol Interaction Microphysics) is a new multi moment microphysics scheme for the UM that includes the effects of aerosol upon clouds and vice versa. This provides enhanced capability over the old operational scheme in which the cloud droplet concentration was constant throughout the domain.

As with other bulk microphysics schemes, the cloud and rain water are separated into two hydrometeor classes. In each class the drop size distributions are describe using a gamma distribution with a prescribed shape parameter and prognosed bulk mass and number concentration, i.e. double moment cloud and rain (for details on the multi-moment implementation see Shipway and Hill, 2012). In this study, ice microphysics is not switched on since only warm clouds were present in the study area.

    If a model grid-box is deemed to be sufficiently humid by the above–mentioned cloud scheme then cloud water condenses

and the number of droplets activated is determined using the scheme described in Abdul-Razzak and Ghan (2000) that makes use of explicitly resolved vertical velocity, humidity and aerosol properties to compute the number concentration of droplets activated. Autoconversion of cloud droplets to rain and droplet accretion is based upon Khairoutdinov and Kogan (2000) and the self–collection of rain follows Beheng (1994). Details on the testing of the warm rain microphysics parameterizations used in CASIM in an idealized framework can be found in Hill et al. (2015). The scheme includes an option for the sedimentation

of cloud water; however, this is switched off for most of the runs in this paper. We discuss the effect of switching this on for some test runs in Section 4.2. The hydrometeor fall–speed relationship follows Shipway and Hill (2012). Table 2 summarizes the microphysical parameterizations used and Table 3 gives the constants used.

    Five different size modes are available to represent soluble and insoluble aerosol, but only a single soluble accumulation mode is used here. The aerosol mode has a lognormal size distribution with a fixed width. In this paper the aerosol is initially

spatially uniform in both the vertical and horizontal and the same aerosol profiles are applied as lateral boundary conditions to the inner nest. There are no local sources of aerosol at present. However, aerosol is advected and thus concentrations can change locally due to convergence and divergence. Details of the aerosol concentrations used in the different runs of this work are given in the next section. CASIM includes the option of "aerosol processing", which includes activation scavenging; in-cloud mechanical processing into fewer, but larger aerosol particles (via collision coalesence); precipitation washout of both

in-cloud and out-of-cloud aerosol, and evaporative regeneration. These processes can lead to an overall reduction in the aerosol available for forming cloud droplets. However, aerosol processing is not switched on for the runs in this work, but will be considered in a later paper.

### 2.1.3 Details on model runs and sensitivities

We have performed several model runs that are listed in Table 4. The run denoted as Old-mphys uses the old microphysics

scheme (Wilson and Ballard, 1999), which also uses the Smith (1990) sub-grid cloud scheme and has a fixed cloud droplet concentration of $100 \ cm^{-3}$. All of the other simulations use the CASIM microphysics. CASIM-Ndvar is the control aerosol case, where the accumulation soluble mode aerosol has been chosen (the mass mixing ratio was set to $4.6 \times 10^{-8} \ kgkg^{-1}$, the number concentration to $3.8 \times 10^9 \ kg^{-1}$) to produce droplet concentrations that are in approximate agreement with those



observed (see Section 3.2.1). CASIM-Ndvar-RHcrit0.999 is the same as the control run except that the sub-grid cloud scheme has been switched off in order to investigate its importance.

Aerosol sensitivity runs have been performed where the soluble accumulation mode aerosol mass and number have been reduced by factors of 10 and 40 (CASIM-Ndvar-0.1 and CASIM-Ndvar-0.025, respectively), and increased by a factor of 10 (CASIM-Ndvar-10). This range of aerosol concentrations creates clouds with droplet numbers that bracket the range observed during the VOCALS field campaign, as we will show in Section 3.2.1.

## 2.2   Observations

Data from a variety of instruments onboard several observational platforms including satellite, ship and aircraft have been used to validate the model. The data used (including error estimates from the literature) are described in Appendix B and summarized in Table 5.

## 2.3   Cloud fraction definition

In this paper we choose to define cloud using an LWP threshold of 20 $gm^{-2}$. The use of LWP makes comparisons between models and satellites instruments simpler. A threshold value of 20 $gm^{-2}$ represents a conservative estimate of the lower limit of the microwave instruments used to observe LWP.

## 3   Results

### 3.1   General case study features from the observations

Figure 2 shows snapshot satellite images from 13th November, including daytime maps of LWP and $N_d$ from GOES-10 and a nighttime LWP map from AMSR-E. Both LWP images reveal extensive cloud cover, although it is evident that there are more cloud free regions in the daytime image. The LWP is much larger at night compared to the daytime (note the different colourbars), which is a well-known feature of the diurnal cycle of stratocumulus and is due to the lack of shortwave heating of cloud tops at night (Wood, 2012). Both the daytime and nighttime plots show that the highest LWP region lies in a NW to SE oriented diagonal band across the region with thin cloud present both near the coast and much further offshore to the southwest. The model domain (indicated by the white box) contains both the coastal thin cloud region and the higher LWP values offshore. The high resolution daytime GOES-10 image shows several cloud-free regions within the general area of the "diagonal band" (e.g. centred at 16º S, 88º W; 19º S, 81º W; and 22º S, 77º W), which do not appear to be present at night. These regions could be considered as POCs ("Pockets of Open Cells") since they constitute large regions of low cloud fraction open cells in amongst a region of otherwise closed cell stratocumulus.

The $N_d$ map shows the presence of a large spatial gradient with high $N_d$ values near the coast and low $N_d$ values offshore, which seem somewhat anticorrelated with LWP. This may indicate correlations caused by meteorology (e.g. two separate airmasses), or it could be the result of aerosol feedbacks upon LWP (or a combination of the two). This boundary crosses the



UM model domain splitting it roughly into two halves in terms of LWP with a low LWP/high $N_d$ region to the NE and a high LWP/low $N_d$ region in the SW.

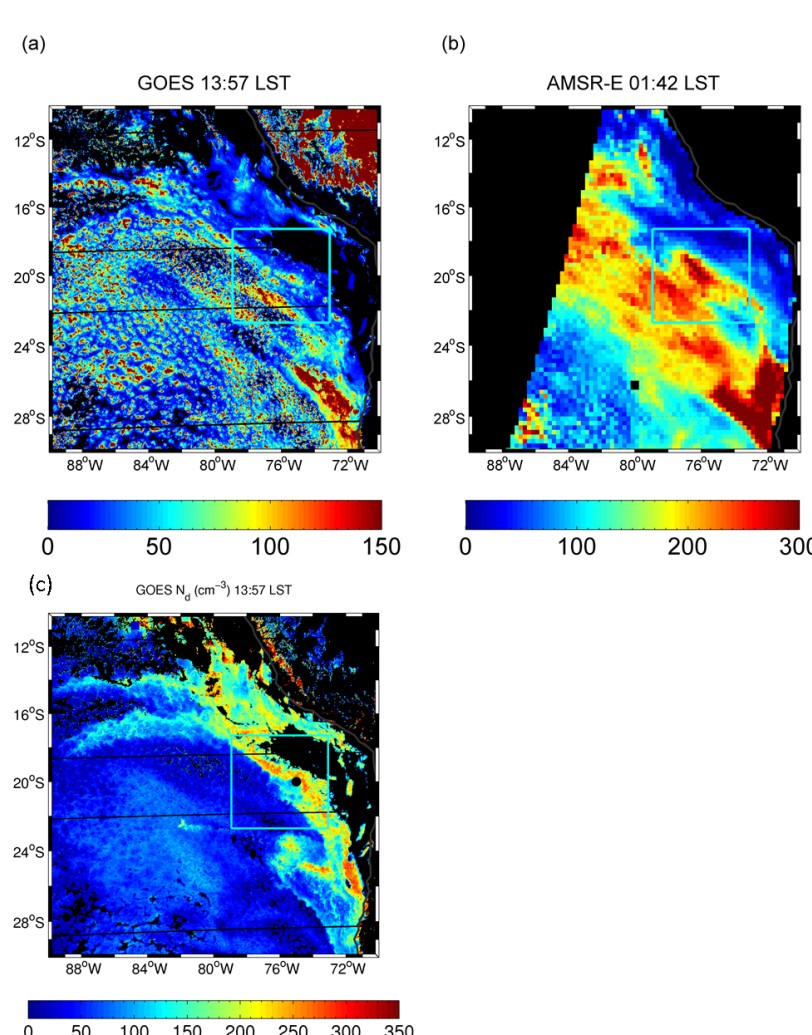

**Figure 2.** Snapshots of LWP (a and b, $gm^{-2}$) and $N_d$ (c) for 13th Nov, 2008. (a) and (c) 13:57 LST (daytime, 18:45 UTC) from the GOES-10 geostationary satellite at 4 km resolution. (b) 01:42 LST (nighttime, 06:30 UTC) from the AMSR-E instrument that has a lower resolution of $0.25^{\circ}$ (GOES-10 LWP and $N_d$ retrievals are probably not reliable at nighttime). Note the different colour scales. The AMSR-E image has a region missing to the west due to the polar orbiting nature of the satellite and the limited swath width. Black regions in the LWP plots denote those where LWP$<20$ $gm^{-2}$, which we have chosen to define as cloud-free. The white box shows the location of the 1 km resolution model domain.



## 3.2 Model validation and aerosol sensitivity

### 3.2.1 Droplet concentration distributions

Figure 3 shows PDFs of the cloud droplet number concentration for a snapshot daytime period (14:00 LST) for the inner nest of the model domain, for both the model and GOES-10 satellite instrument. Since the satellite provides a 2D field of $N_d$, it is

necessary to make a 2D field from the 3D model data. This is done by taking $N_d$ at the height of the maximum LWC with each model profile, since this helps to avoid the $N_d$ from spuriously small LWC gridboxes from being included. For similar reasons, datapoints from both the model and the satellite are ignored if the LWP is less than 5 $gm^{-2}$, but only after the model $N_d$ and LWP data has been coarse grained from its native 1 km resolution to that of GOES-10 (4 km).

   The observations from GOES-10 show that there is a two mode PDF, with a mode of very low $N_d$ ($\sim$25 $cm^{-3}$) and one

at 215 $cm^{-3}$. This is reflecting the two different airmasses that seem to be present, as discussed earlier (Fig. 2, i.e. a near-coastal airmass with high $N_d$ and an offshore airmass with low $N_d$). The models only captures one of these modes of $N_d$ since a spatially uniform aerosol field was applied. However, it is conceivable that the low $N_d$ mode may also be the result of aerosol removal within the precipitating open cell regions of the stratocumulus since this too can lead to very low $N_d$ values (B13). Since aerosol processing and scavenging is not switched on for these runs the model will not capture the latter process.

Using fixed aerosol concentrations allows the exploration of the extreme high and low aerosol loading scenarios without the complications of aerosol source functions and processing. This extra complexity will be explored in a later paper.

   The control model (CASIM-Ndvar) has a $N_d$ distribution that has a similar width (to within $\sim$15%) to that of the higher $N_d$ mode observed by GOES-10, although it has a higher frequency of the higher $N_d$ values (above 275 $cm^{-3}$). Despite this, the modal value is lower for the model (155 $cm^{-3}$) than for the large mode of GOES-10 (215 $cm^{-3}$), although the broadness of

the model distribution means that it still has large frequencies of data at the position of the GOES-10 modal value. Given the lack of sensitivity of the modelled clouds to increasing the aerosol by a factor of ten described shortly, it seems unlikely that the small differences between the modelled and observed large $N_d$ mode would have a very large impact on cloud properties. The lack of a lower $N_d$ mode in the model could be more important. This is explored through the sensitivity tests where we reduce the aerosol.

Figure 3 demonstrates that by reducing the aerosol by factors of 10 and 40 (CASIM-Ndvar-0.1 and CASIM-Ndvar-0.025) decreases the mode values of $N_d$ to 25 and 3.5 $cm^{-3}$, respectively. The CASIM-Ndvar-0.025 case produces droplet concentrations that are very low with no values above 10 $cm^{-3}$. This is consistent with the observations of ultra-clean regions that have been observed in the outflow regions of POCs (Wood et al., 2011a) and so can be considered as a lower realistic bound for aerosol concentrations. The CASIM-Ndvar-10 case produces droplet concentrations of up to around 3000 $cm^{-3}$, although

with a 95th percentile of 1585 $cm^{-3}$. Aircraft observations from the VOCALS field campaign reported maximum $N_d$ values of around 400 $cm^{-3}$ in the vicinity of the coast at 20° S (Zheng et al., 2011) over the whole campaign period and so the modelled $N_d$ values in the CASIM-Ndvar-10 case are somewhat higher than those likely to occur in reality for this region. However, $N_d$ values as high as those from the model have been observed elsewhere, for example within stratocumulus over the East China Sea (Koike et al., 2012) and so this simulation represents the upper bound of $N_d$ values that are likely to occur anywhere on





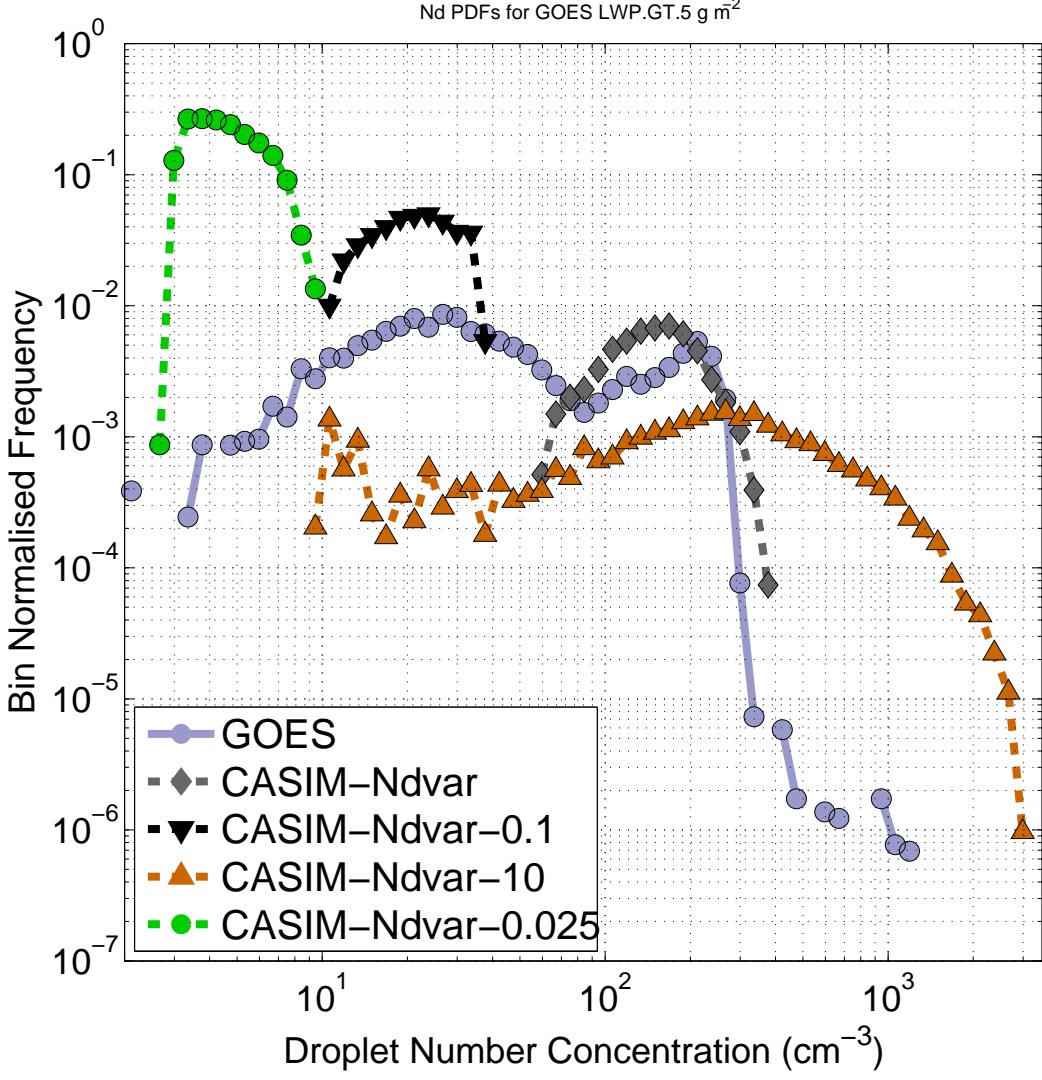

**Figure 3.** PDFs of the cloud droplet number concentration for the model domain region. A snapshot time of 14:12 LST is used for the model and 13:57 LST for the GOES-10 satellite, which is the nearest available data point. 3D model data is first converted to 2D data taking the $N_d$ at the height of the maximum LWC with each model profile. The model data is subsequently coarse grained from its native 1 km resolution to that of GOES-10 (4 km). Datapoints from both the model and the satellite are ignored if the LWP is less than 5 $gm^{-2}$.

earth. We will show later (e.g. Sections 3.2.2 and 4.2) that the exact value for the upper bound of aerosol concentrations is not important given the lack of impact of aerosol on cloud properties as demonstrated by comparisons between the CASIM-Ndvar and CASIM-Ndvar-10 cases.




### 3.2.2 LWP and RWP timeseries

Figure 4 shows a timeseries of the mean LWP over the region of the UM domain for the different model simulations and the satellite observations. For the latter, both microwave instruments and GOES-10 retrievals are shown. There are several microwave instruments that give snapshots throughout the diurnal cycle. GOES-10 data is only used for the daytime, but

5   gives higher time resolution. During the daytime, GOES-10 and the microwave instruments agree within $\sim$10 $gm^{-2}$ giving confidence in the observations.

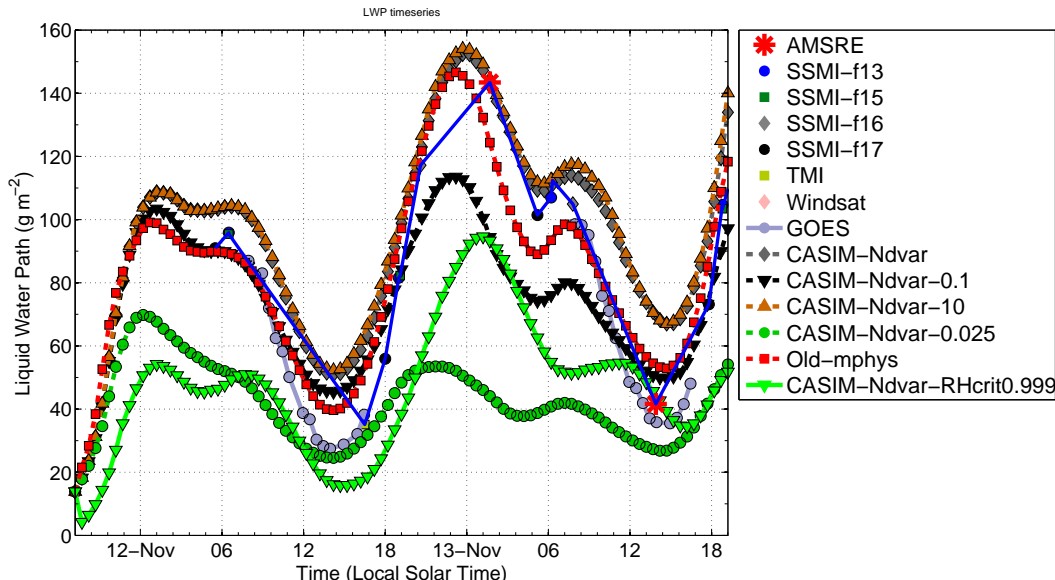

**Figure 4.** Timeseries of the mean LWP over the region of the UM domain for the different model simulations, the microwave satellite instruments and the GOES-10 instrument. There are several microwave instruments that give snapshots throughout the diurnal cycle, as labelled in the legend; they are joined by the blue line. GOES-10 data is only used for the daytime, but gives higher time resolution and retrievals where the solar zenith angle is larger than 65$^o$ have not been included due to likelihood of biases as detailed in (Grosvenor and Wood, 2014).

The model runs produce the observed peaks and troughs in LWP and even capture the secondary peak on 13th Nov, at around 8 LST. The higher aerosol runs (CASIM-Ndvar and CASIM-Ndvar-10) and the old microphysics run (Old-mphys) also capture the magnitude of the LWP values well, although all simulations overestimate the daytime LWP values. There is

10   better agreement for Old-mphys (overestimate of around 10 $gm^{-2}$, or 10%) than for the CASIM runs (overestimate of around 20-30 $gm^{-2}$, or 50–75%; although note that the observed LWP is low at this time, being only 40 $gm^{-2}$, resulting a large percentage bias). The reverse is true for nighttime values where the CASIM runs match the observations very well (within 10 $gm^{-2}$, or 10%), but the Old-mphys run underestimates by 15–20 $gm^{-2}$, or 15–20%.





In the lower aerosol runs (CASIM-Ndvar-0.1 and CASIM-Ndvar-0.025) LWP values are significantly lower, indicating a cloud macrophysical response via the precipitation rate. For the high aerosol case little impact of aerosol on the cloud field was found relative to the control case (CASIM-Ndvar). This is because there is little rain production occurring in the control case and hence the addition of more aerosol cannot have much of a precipitation suppression impact. This is demonstrated in Fig 5 where rain water path (RWP) is between 8 and 11 times lower in the control case than in the lowest aerosol case.

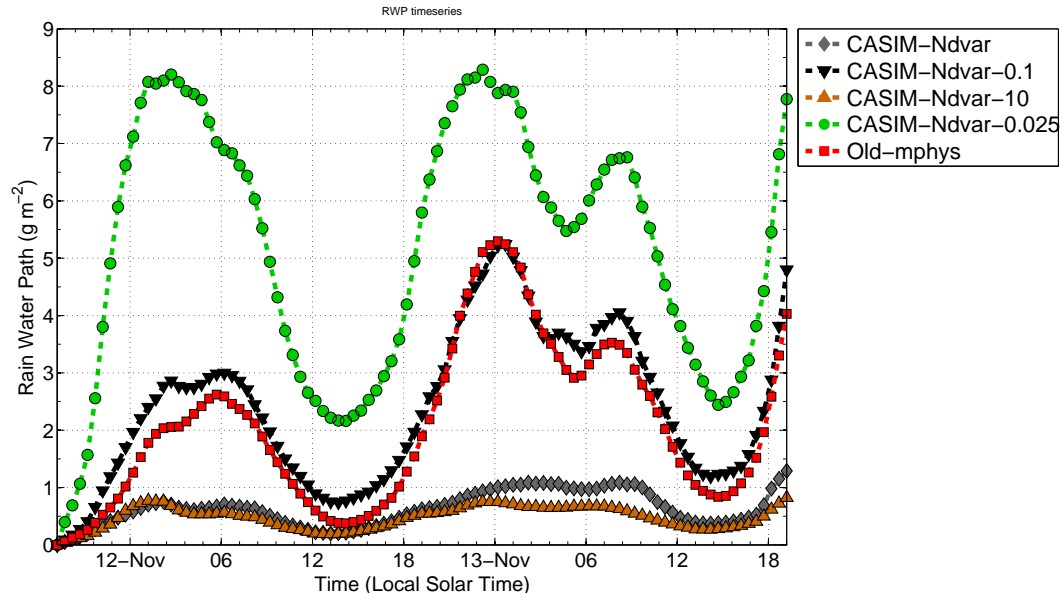

**Figure 5.** As for Fig. 4, except for RWP and for the models only.





CASIM-Ndvar-RHcrit0.999 is a model run where the sub-grid cloud scheme has been switched off, which results in a very large LWP reduction compared to the control case with LWP values similar to those from CASIM-Ndvar-0.025 for the first day and CASIM-Ndvar-0.1 for the second day. The results clearly highlight that while it is possible for the aerosol environment to have a large impact on the structure of the stratocumulus cloud deck (cf CASIM-Ndvar-0.025 and CASIM-Ndvar), the role of

the treatment of subgrid humidity, even for grid spacings of 1km, is still as important as a factor of 10-40 reduction in aerosol loading (cf. CASIM-Ndvar and CASIM-Ndvar-0.025 or CASIM-Ndvar-0.1). Given the unrealistically low LWP values in the CASIM-Ndvar-RHcrit0.999 case the results from this run will not be included in future plots for clarity.

### 3.2.3   LWP maps and cloud coverage

Fig. 6 shows the same daytime satellite LWP image from the GOES-10 satellite that was shown in Fig 2, but zoomed in
to the region of the model domain. Also shown are corresponding images from the control, very low and high aerosol runs (CASIM-Ndvar, CASIM-Ndvar-0.025 and CASIM-Ndvar-10). The satellite image reveals that the clouds are orientated in diagonal band-like structures of high LWP and also shows the structure of the POC regions; i.e. small regions of higher LWP (presumably the updraft region) surrounded by regions of negligible cloud (downdraft/cold pool front region). There are two main POC regions within the model domain region; one centred at around 19.75º S, 78.25º W and another more elongated
region centred at 22.5º S, 76.75º W, but stretching to the NW and SE. Hereafter, these will be referred to as the upper and low POC regions, respectively.

The control and high aerosol simulations qualitatively represent the diagonal band structures and the low LWP values near the coast (in the NE corner of the domain) very well, despite the fact that there is no spatial gradient in the aerosol field of the model, as there would be in reality. This indicates a general dominance of the meteorological state upon the macrophysical
properties of the clouds. However, the simulations where the aerosol amount is changed show a dramatic sensitivity when decreasing the aerosol from the control level by a factor of 40. In the very low aerosol run (CASIM-Ndvar-0.025) LWP values and the cloud fraction are significantly lower, indicating a cloud macrophysical response via the precipitation rate, similar to what has been observed in LES studies (Berner et al., 2013; Feingold et al., 2015). For the high aerosol case little impact of aerosol on the cloud field was found relative to the control case. This is because precipitation is low in the control case (Fig. 5)
and so the addition of more aerosol cannot influence precipitation.

In the region to the south of where the upper POC is situated in reality (between the bands of cloud) there is a general region of more broken cloud in CASIM-Ndvar and CASIM-Ndvar-10 cases. However, the scale of the convective cells in these runs is much larger than that in the real POC and also the lower POC region is not captured. The low aerosol run produces small–sized convective cells surrounded by clear air that are reminiscent of the observed POC regions, but they occur throughout the whole
domain. Thus, this suggests that the model is capable of producing open-cell features given low enough aerosol concentrations, but cannot reproduce isolated POC regions in amongst the closed cell convection.

The nighttime LWP maps in Fig. 7 show that a large area of high LWP cloud is observed by AMSR-E. The CASIM-Ndvar and CASIM-Ndvar-10 models also produce a large region of high LWP cloud in the SW corner of the domain, but this is less widespread than observed and reaches higher LWP values. As for the daytime, there is little response between the CASIM-





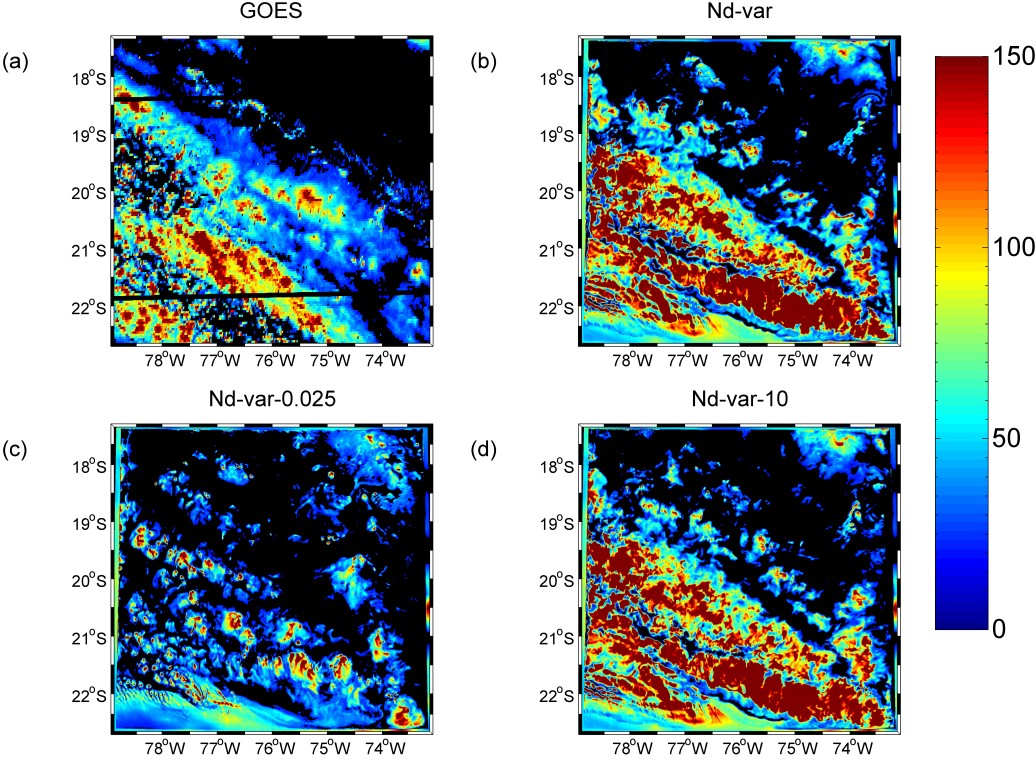

**Figure 6.** Daytime snapshots of LWP ($gm^{-2}$) for 13th Nov, 2008 for the region of the inner model domain. (a) GOES-10 satellite at 13:57 LST (daytime); (b) model with control aerosol; (c) model with x0.025 aerosol; (d) model with x10 aerosol. The model images are from 14:12 LST. Regions where the LWP is less than 20 $gm^{-2}$ are plotted as black to give an estimate of cloud fraction.

Ndvar and CASIM-Ndvar-10 cases, but a large response when aerosol is reduced (CASIM-Ndvar-0.025), again with lower LWPs and lower cloud fractions that are similar to open–cell stratocumulus.

### 3.2.4 LWP distributions

Figure 8 shows PDFs of LWP from the model and satellite for a daytime snapshot at around 14:00 LST. Both the AMSR-E and the GOES-10 satellite are shown, and the model and GOES-10 data have been coarse grained to the AMSR-E resolution of 0.25°. GOES-10 and AMSR-E generally agree to within 20 $gm^{-2}$ indicating only a small amount of observational uncertainty. The control (CASIM-Ndvar) and high aerosol (CASIM-Ndvar-10) cases show some agreement with the observations, although frequencies are too low for LWP values between 40 and 80 $gm^{-2}$ (CASIM-Ndvar underestimates the bin normalised frequency



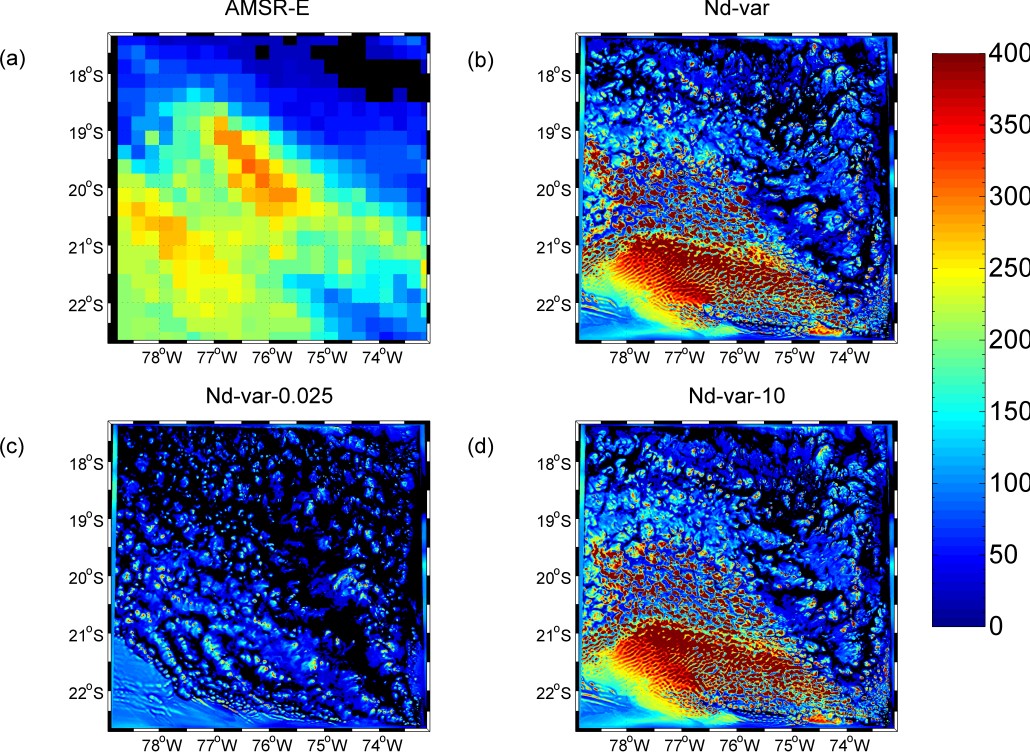

**Figure 7.** As for Fig. 6 except for nighttime (01:42 LST for AMSR-E and 02:12 LST for the model).

for the bin centred at 50 $gm^{-2}$ by 22–55% and CASIM-Ndvar-10 by 44–67%, respectively[1]) and too high for LWP values >∼110 $gm^{-2}$ (AMSR-E had between 3 and 5% of the data within this range compared to 19-23% for CASIM-Ndvar, and 18-22% for CASIM-Ndvar-10), which is consistent with the overprediction of the mean LWP (Fig.4). The CASIM-Ndvar-0.1 and Old-mphys runs are very similar and also exhibit overprediction of high LWP values, but to a lesser degree. The CASIM-
5  Ndvar-0.025 run has too few high LWP points and too many low ones, again consistent with the underprediction of mean LWP for this run compared to the observations.

Also shown are nighttime LWP PDFs from the AMSR-E satellite only. The observations show a peak at around 230 $gm^{-2}$, which is not captured by the models. For example, the CASIM-Ndvar and CASIM-Ndvar-10 cases underestimate the frequencies of the bin at LWP=230 $gm^{-2}$ by ∼50–70%. Instead the models have frequencies that are too high at lower LWPs of
10  around 30–100 $gm^{-2}$, but with the CASIM-Ndvar and CASIM-Ndvar-10 runs showing slightly less overprediction than the other runs (∼ 70–200% overestimate for the latter runs). As demonstrated for the daytime, the CASIM-Ndvar-0.025 run has a

---

[1]Here and elsewhere in this section, the minimum and maximum of the discrepancy between the models and observations are estimated using Poisson counting statistics for the bin counts in the PDFs



(a)

(b)

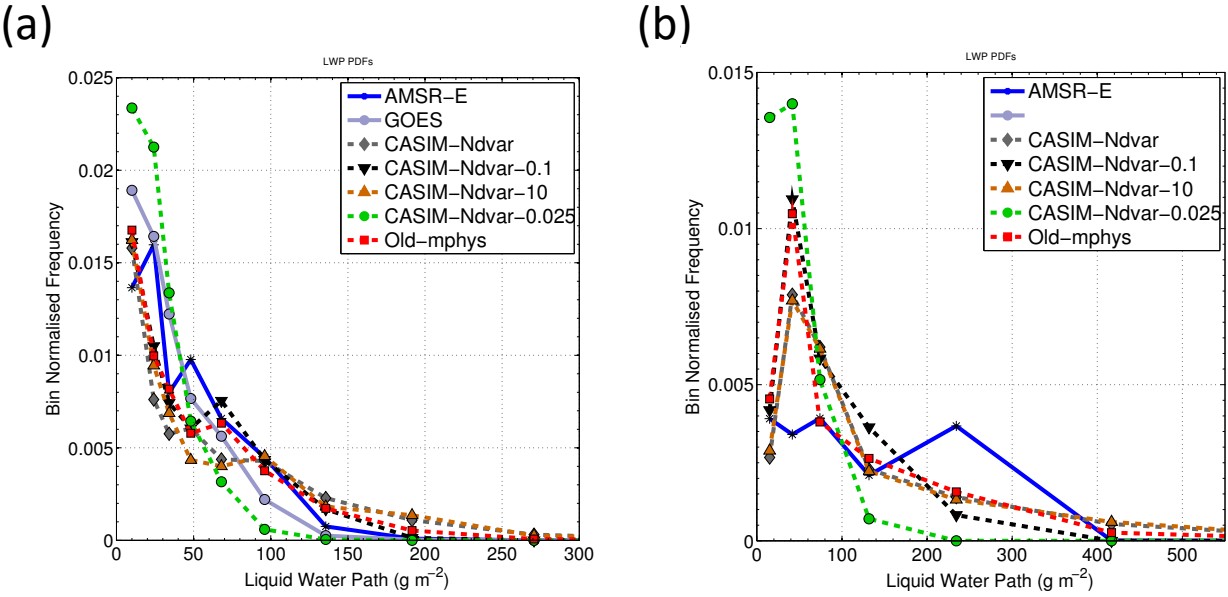

**Figure 8.** PDFs of LWP for daytime (left) and nighttime (right) snapshots on 13th November for the model and for satellite observations. For the daytime, a snapshot time of 14:12 LST is used for the model, 13:57 LST for the GOES-10 satellite and 13:54 LST for AMSR-E. For the nighttime, only the AMSR-E satellite is shown with a time of 01:42 LST. The model time is 02:12 LST. The model and GOES-10 data have been coarse grained to the AMSR-E resolution of 0.25°.

much larger number of low LWP values compared to the observations and the other models indicating excessive LWP removal by precipitation. Consistent with this, the higher aerosol runs (CASIM-Ndvar and CASIM-Ndvar-10) and the Old-mphys run have a small number of points with LWP larger than around 300 $gm^{-2}$ (between 11 and 14 % of the total counts according to Poisson statistics for the CASIM-Ndvar case), which are not present for the low aerosol runs (CASIM-Ndvar-0.025 and

5  CASIM-Ndvar-0.1). There are also higher frequencies of the higher LWP values in the CASIM-Ndvar-0.1 case compared to the CASIM-Ndvar-0.025 case (e.g. between 9 and 12 % of total counts have LWP>170 $gm^{-2}$ for CASIM-Ndvar-0.1, but none for CASIM-Ndvar-0.025). Since this is the second day of the simulation and precipitation has been occurring throughout the





simulation in the low aerosol runs, there will have been some degree of conditioning (drying) that has occurred before the time shown here.

Fig. 7 showed that the CASIM-Ndvar and CASIM-Ndvar-10 runs failed to capture the high areal coverage cloud of moderate LWP that was observed. They instead featured a smaller region of higher LWP with the rest of the domain consisting of lower
LWP cloud. All of these features are evident in the LWP PDFs. The results suggest the failure of the model to capture the large area of moderate LWP cloud is unlikely to be due to microphysical processes related to excess rain formation for the CASIM-Ndvar and CASIM-Ndvar-10 cases since very little precipitation is occurring in these runs. It seems more likely that this problem stems from a combination of the sub-grid cloud scheme (the choice of sub-grid humidity variables for CASIM was untuned), the boundary layer scheme, or the large scale meteorological fields coming from the global model.

In addition to spatial satellite PDFs the microwave radiometer onboard the RH Brown ship can provide a measure of temporal variability at a resolution of 10 minutes, but at a fixed location (20º S, 75º W). Figure 9 shows LWP PDFs from the ship and for various model runs for the time period 06 UTC (01:12 LST) 12th Nov to 0 UTC on 14th November (19:12 LST on 13th Nov). A long sampling period is necessary given the restriction of the ship sampling being at only one location.

Figure 9 shows that all of the models underestimate the occurrence of LWP$> 150 \ gm^{-2}$ values and overestimate the occur-
rence of the lower LWP values for this region (LWP$<50$–$100 \ gm^{-2}$ depending on the model run). However, the overestimate of the low values is much less severe for the CASIM-Ndvar and CASIM-Ndvar-10 runs. These results are consistent with the AMSR-E nighttime results from Fig. 8 suggesting that the issues highlighted with the snapshot nighttime map (Fig. 7) are pertinent for the whole simulation, at least for the region of the ship.

### 3.2.5 Distributions of cloud fraction

Figure 10 shows distributions of cloud fraction ($f_c$) for the model runs and for the GOES-10 satellite for the whole of the daytime period of 13th November (between approximately 08 and 16 LST). For the cloud fraction calculation, the model LWP data is first coarse-grained to the GOES-10 resolution of 4 km and then the cloud fraction is defined within 0.25º regions based on a LWP threshold of $20 \ gm^{-2}$.

Figure 10 shows agreement between the model and observations to within 0.1 in terms of cloud fraction for all of the runs,
except for the lowest aerosol case (CASIM-Ndvar-0.025) suggesting that the model is very capable of correctly simulating the balance of cloudy and cloud-free areal coverage. For the model runs (excluding the lowest aerosol case) and the observations the $f_c$ bin that contributes most to the frequency is the highest $f_c$ value ($>0.95$) corresponding to nearly overcast conditions. The Old-mphys and CASIM-Ndvar-0.1 runs both have similar contributions to the frequency from this bin, as well as for other $f_c$ values at the upper end of the distribution, with both underestimating the contribution to the total compared to the
observations (i.e. these models do not have enough of the higher $f_c$ values). In contrast the CASIM-Ndvar and CASIM-Ndvar-10 agree with the observations to within $\sim$0.04 cloud fraction for $f_c>0.7$. Thus, CASIM microphysics with the sub–grid cloud scheme seems to represent an improvement over the old microphysics in this aspect.

The low aerosol case (CASIM-Ndvar-0.025) showed a much greater frequency of the lower cloud fractions and very few fully overcast datapoints, which is consistent with the results and discussion of the snapshot maps (discussed above).




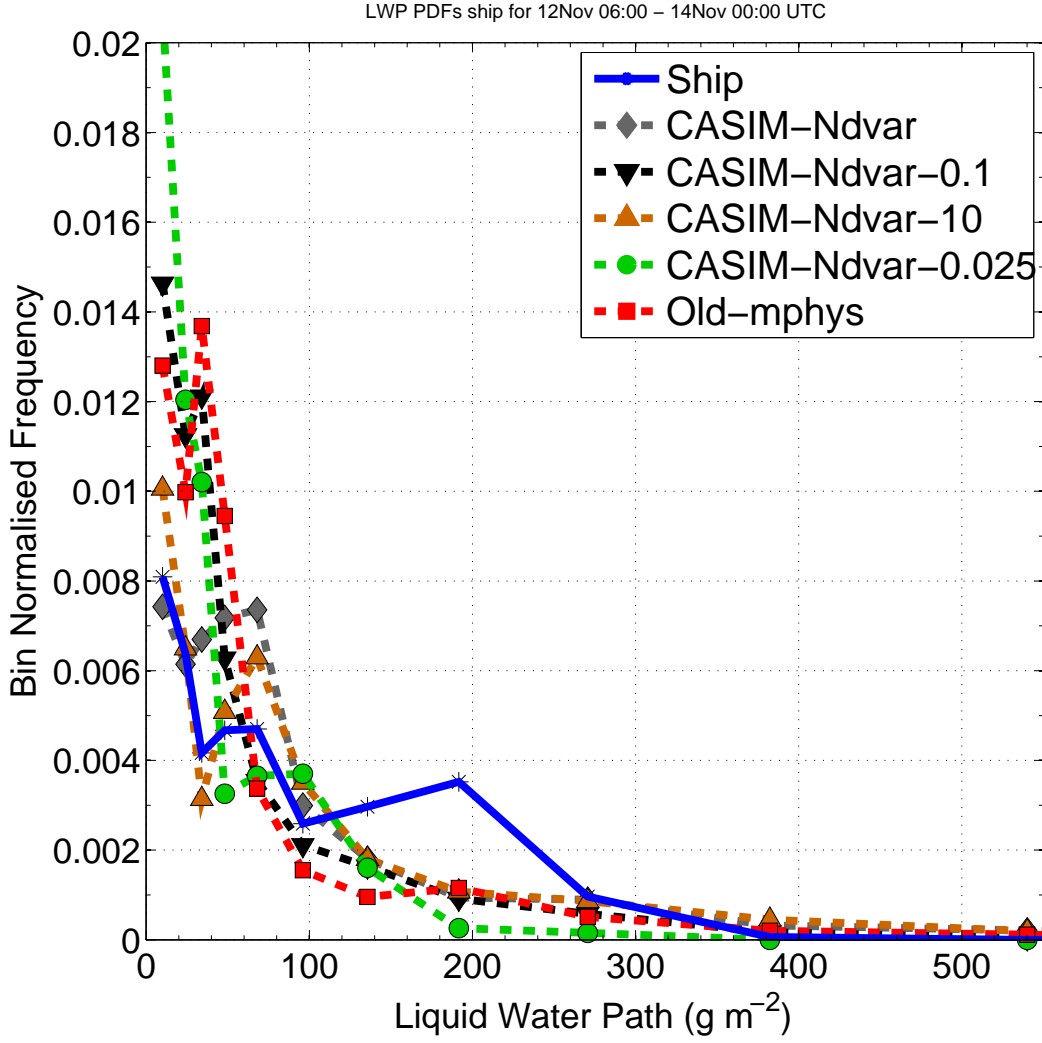

**Figure 9.** PDFs of LWP for the period from 6 UTC on 12th November to 0 UTC on 14th November, 2008 from the RHB ship microwave radiometer and various UM model runs. 10 min averaged data is used for both the ship and the model. For the model the $3\times3$ gridboxes centred at the location of the ship ($20^o$ S, $75^o$ W) are used and combined into one PDF in order to account for the possibilty of spatial variability.

A distribution plot is not possible for the nighttime where only coarse resolution ($0.25^o$) snapshots from the microwave instruments are available. The domain average cloud fraction using the same 20 $gm^{-2}$ LWP threshold was 0.95 for the AMSR-E satellite image (Fig. 7), which is very similar to that predicted in the control and high aerosol cases (0.97 and 0.98, respectively after coarse graining to $0.25^o$). Again, the very low aerosol case (CASIM-Ndvar-0.025) showed the tendency to





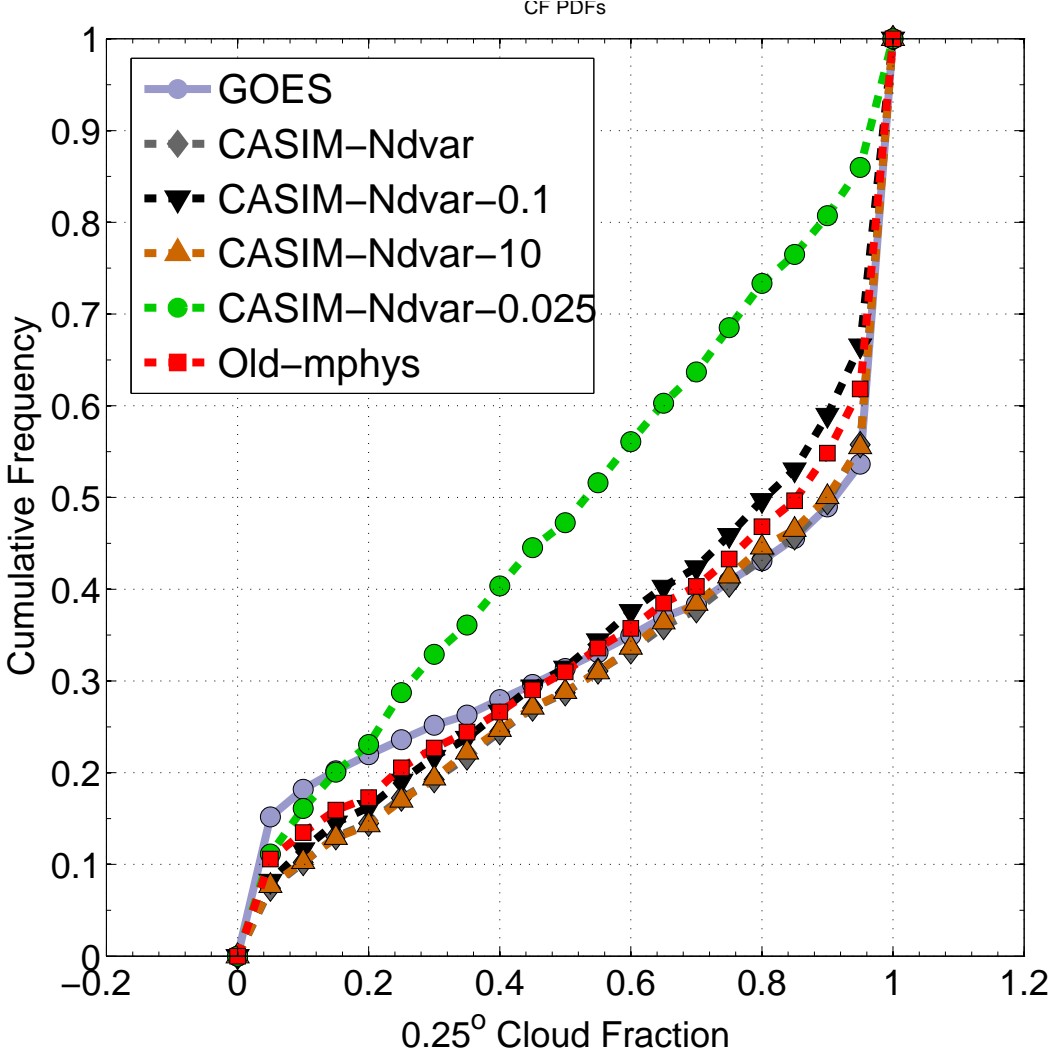

**Figure 10.** Cloud fraction cumulative distribution functions between 08:12 and 16:12 LST for the model and 07:57 and 15:57 for GOES-10 sampled every 30 minutes. Here each cloud fraction value is calculated as the fraction of datapoints with LWP greater than 20 $gm^{-2}$ relative to the total number of datapoints in 0.25° × 0.25° areas. GOES-10 data is used at the native 4 km resolution and the model LWP data is first coarse grained to this resolution from the native 1 km resolution.

produce a much lower cloud fraction (0.79). Thus, the model is showing large sensitivity of the cloud coverage to aerosol for the whole of the daytime period and likely for the nighttime period too.





### 3.2.6  Radiative flux distributions

Figure 11 shows PDFs of the shortwave and longwave top of the atmosphere (TOA) fluxes from CERES and the models for the region of the model domain (hereafter $SW_{upTOA}$ and $LW_{upTOA}$, respectively). The results show that the aerosol amount has a large influence on the SW fluxes with the CASIM-Ndvar-0.1 and CASIM-Ndvar-0.025 runs showing mode fluxes that

are much lower than those observed. The other runs produce SW distributions that are very similar to the those observed with CERES. The low values in the CASIM-Ndvar-0.1 run are unlikely to be due to cloud fraction differences since Fig. 10 showed similar distributions for the this and higher aerosol runs. Instead, the response of the SW flux results from the marked change in LWP when aerosol is changed (Fig. 4) and from the change in $N_d$ (Fig. 3); this attribution is discussed in more detail in Section 4.3. There is a shift in the LW flux towards higher fluxes relative to the control case for CASIM-Ndvar-0.025, but not

for CASIM-Ndvar-0.1. This shift is due to the cloud fraction response seen in CASIM-Ndvar-0.025, but not in CASIM-Ndvar-0.1 (Fig. 10); a lower cloud fraction will mean larger LW fluxes due to CERES detecting radiation emitted from more surface points with correspondingly warmer temperatures.

The CASIM-Ndvar, CASIM-Ndvar-10 and Old-mphys runs all produce SW and LW distributions that are relatively close to those observed. There are a few discrepancies such as the observed peak in SW frequencies between 500 and 700 $Wm^{-2}$

not being captured by the models, which show correspondingly larger peaks at lower SW values. PDFs of LWP and $N_d$ at the specifc times of the SW CERES overpasses for the model and GOES-10 (not shown) suggest that the error is attributable to an underestimate in the number of $N_d$ values between around 210 and 300 $cm^{-3}$. The Old-mphys run has $N_d$ fixed at 100 $cm^{-3}$. For LW the Old-mphys run has values that are shifted slightly towards lower LW values compared to the other runs, which agrees with the observations better than the CASIM runs for the upper tail of the distribution, but not as well for the lower

tail. The modal value of the CASIM runs is also slightly too high compared to the observations, whereas the mode for Old-mphys is slightly too low. The mean values of the CASIM-Ndvar, CASIM-Ndvar-0.1 and CASIM-Ndvar-10 runs agree with CERES within 0.2%, with the Old-mphys and CASIM-Ndvar-0.025 runs performing slightly worse (-0.5 and +1.1% biases, respectively).

Figure 12 shows the equivalent plot, but for nighttime snapshots. The models generally all produce distributions that are

shifted to too high LW flux values, indicating either clouds that are too low in altitude, or cloud fractions that are too low. As for the daytime results, there is a much greater shift to high values for the CASIM-Ndvar-0.025 run indicating the lesser cloud coverage in this case. The Old-mphys is shifted to slightly lower values compared to the other runs as was also the case for the daytime LW fluxes. Again Old-mphys agrees better than the other runs for the upper tail of the observations, but not for the lower tail, which is representative of cloud top conditions.

### 3.2.7  Radar reflectivity

Also onboard the RH Brown ship was a W-band radar, which provided vertical profiles of radar reflectivity (dBZ). This is useful for evaluating the vertical placement of cloud in the model as well as helping to assess the amount of rain formation and its vertical distribution. Fig. 13 shows 2D PDFs of the ship and model radar reflectivity vs height. Model data is used from all



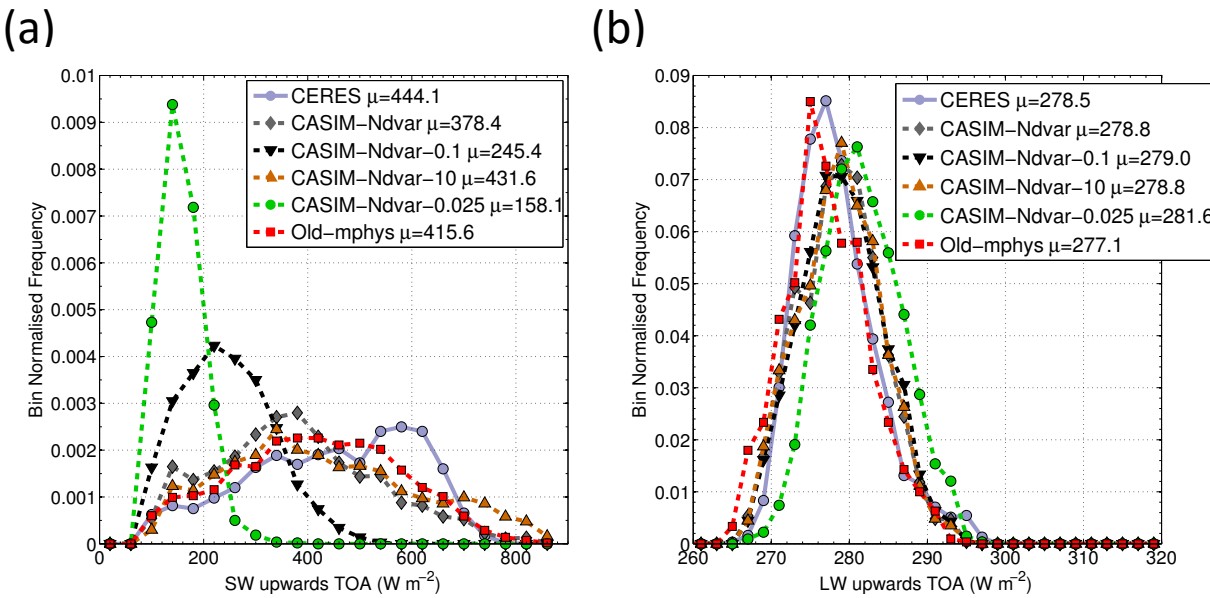

**Figure 11.** Shortwave (left) and longwave (right) top of the atmosphere radiative flux PDFs from CERES and the model for daytime periods for the region of the model domain. This is a combined PDF from the three separate snapshot overpass times of the CERES satellite for the model domain that were available for the simulation period: 10:24 LST (15:12 UTC) on 12th Nov (Terra satellite), 14:19 LST (19:06 UTC) on 12th Nov (Aqua satellite) and 11:07am LST (15:55 UTC) on 13th Nov (Terra). For the model, the three closest available times were used: 15 and 19 UTC on 12th Nov; 16 UTC on 13th Nov. Note, that CERES-Aqua data is not available for the afternoon (local time) of 13th November. The model data was first coarse grained to 20 km, which is the approximate resolution of the CERES data.

of the gridpoints in the $1 \times 1°$ region that surrounds the ship location to compute radar reflectivity based on Rayleigh scattering. The ship radar results show a single mode of reflectivity up to values of around -4 dBZ that likely represents cloud droplets or small drizzle rather than larger rain droplets, since model data in BA12 suggested that rain would appear as a separate mode at higher reflectivity values between around -10 and +10 dBZ. Most of the datapoints lie above 550 m in altitude, although there is some data from lower altitudes and there is a strong mode centred at 1135 m and -24 dBZ. The cloud reflectivity generally





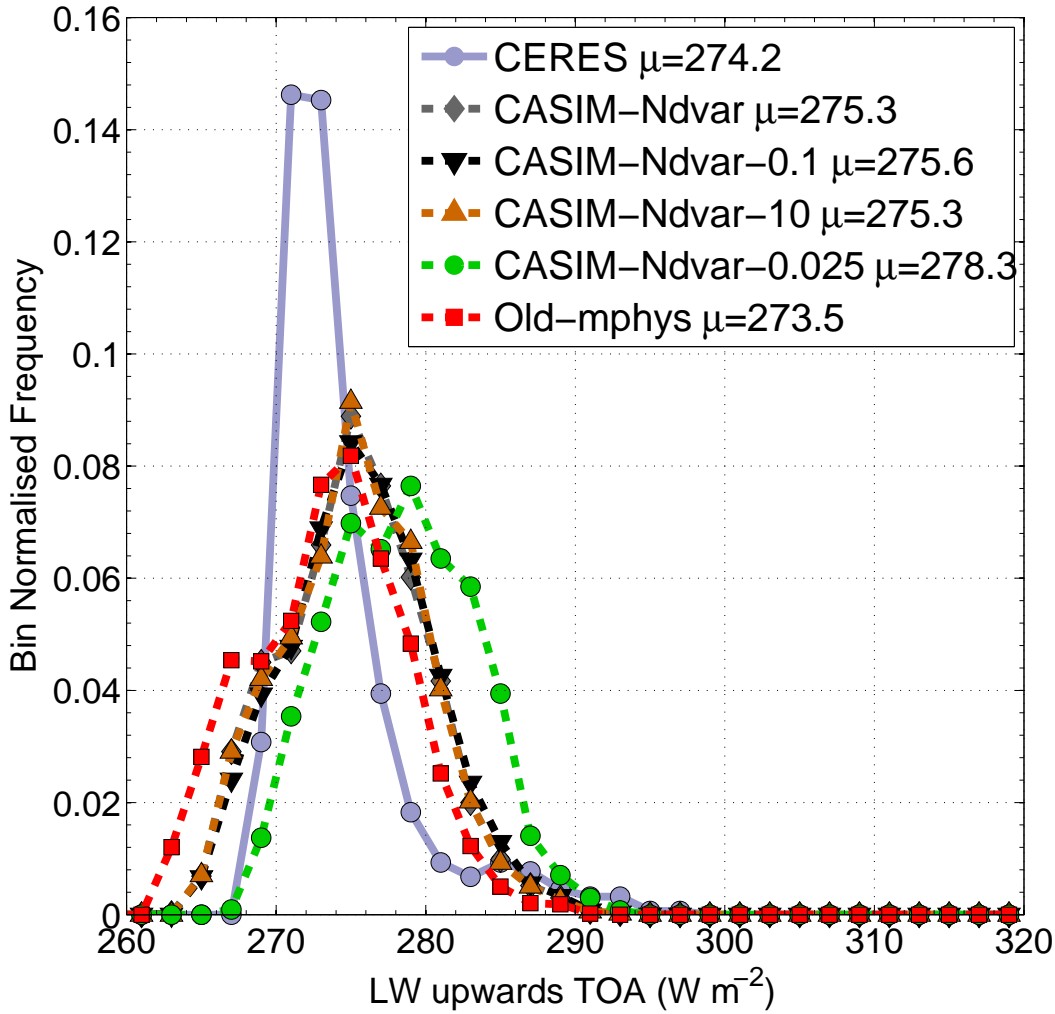

**Figure 12.** As for Fig 11 except for nighttime periods and for longwave only. The times used for CERES are 22:35 LST (03:23 UTC) on 12th Nov (Terra satellite), 02:39 LST (07:27 UTC) on 12th Nov (Aqua satellite) and 01:44am LST (06:32 UTC) on 13th Nov (Aqua). For the model, the three closest available times were used: 03:30 and 07:30 UTC on 12th Nov; 06:30 UTC on 13th Nov.

grows with altitude up to around 1 km, which is consistent with the growth of cloud droplets from above cloud base. Above this height the reflectivity starts to reduce; this could signify the evaporation of cloud due to the entrainment of dry free tropospheric air into the upper regions of the clouds (A04), which could reduce reflectivity through changes in droplet size, number, or both. Alternatively, it may be the result of the presence of less drizzle near the top of the cloud. Most of the observed cloud is below

5   1325 m in altitude, although some cloud was observed up to 1460 m (within the accuracy of the height bins). It is likely that





this corresponds closely with the boundary layer height because cloud tops for stratocumulus generally correspond with the inversion height.

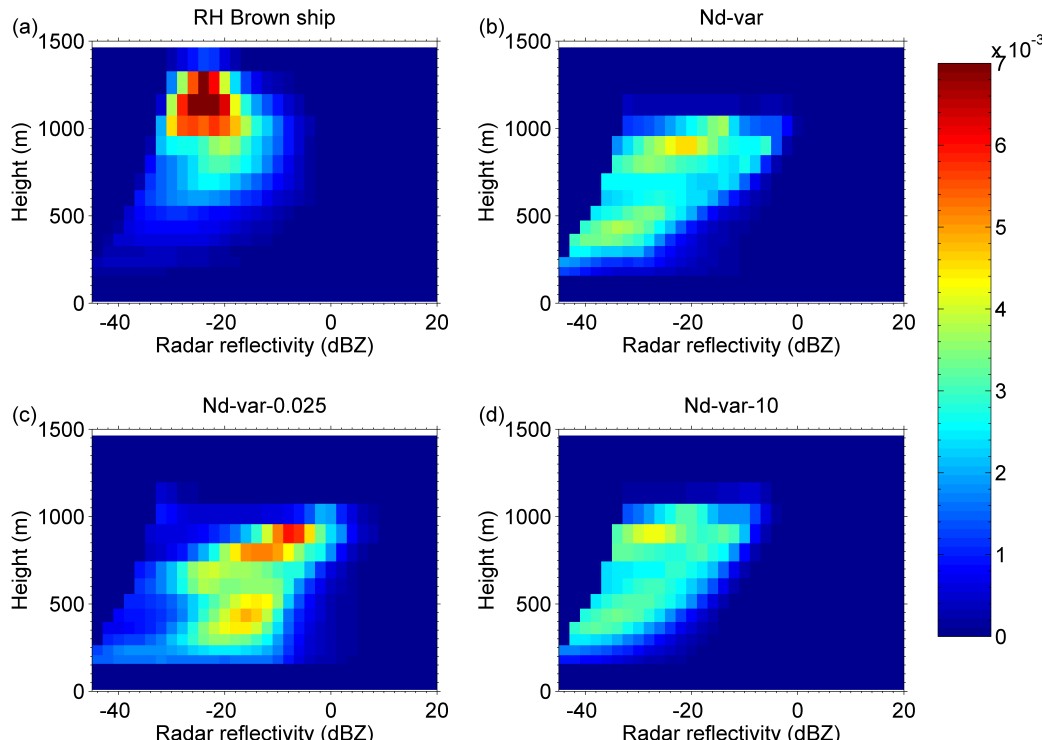

**Figure 13.** 2D relative frequency plot for radar reflectivity vs height from the RH Brown ship W-band radar and for various model runs. The relevant model data is available every 30 minutes and data between the times of 06 UTC (01:12 LST) on 12th Nov and 0 UTC (19:12 LST) on 14th Nov are used, with the first six hours of model data being avoided due to spin-up. Ship radar profiles are available every 0.3 seconds and are used for the same time period as the models. Model data are used for the $1 \times 1^o$ region around the location of the ship. The colours indicate the normalised frequency of occurrence for each bin and the ship and model bin sizes are the same. Note that many of the datapoints lie beyond the minimum value on the x-axis (i.e. very low radar reflectivity) and are not visible, but they are included for the normalisation. The ship radar has a minimum detectable reflectivity that increases with height; model data below this height dependent threshold was set to -1000 dBZ.

The CASIM-Ndvar (control model) and CASIM-Ndvar-10 model results are similar to each other, but show some significant differences compared to the observations. The cloud from the model does not extend much above 1200 m in contrast to the observed cloud reaching 1460 m. This height difference corresponds to approximately two model levels at these altitudes. Also, the model reflectivities do not tend to reduce with height towards the cloud top as they do in reality. The model also has a





higher frequency of points at lower altitudes compared to the observations; e.g. the maximum height of the $3 \times 10^{-3}$ frequency contour is 693 m for the observations and 293 m for the model. This leads to an overall cloud depth (as calculated using the above frequency contour) of 767 m for the observations and 907 m for the model (an 18% overestimate). In the region of the lower altitude cloud in the model there is also a higher frequency of lower reflectivity values in the range -40 to -20 dBZ than

is observed.

However, there are also some similarities; both the models and the observations show a general increase in reflectivity in the lower regions of the clouds with a vertical gradient that is similar to that observed. Also, the highest dBZ values reached (99.9th percentile of all data, including cloud free regions) were around -5.2 dBZ for the observations, whereas for the model the equivalent values were -1.9, -6.7 and +7.4 dBZ for the CASIM-Ndvar, CASIM-Ndvar-10 and CASIM-Ndvar-0.025 runs,

respectively. Thus, the maximum droplet/raindrop embryo sizes reached in the model were close to those in reality for the CASIM-Ndvar and CASIM-Ndvar-10 cases. In the CASIM-Ndvar-0.025 case the low aerosol concentrations are allowing larger droplets to form.

## 4   Discussion

A km-scale regional model using cloud aerosol interacting microphysics has been used to simulate stratocumulus in the SE

Pacific. It was seen that the introduction of the treatment of subgrid humidity (the cloud scheme) was important for simulating the observations. The range of aerosol loading used in the sensitivity studies resulted in droplet concentrations that included the observed range and captured extreme conditions for stratocumulus cloud. This provided insight into the relative importance of cloud brightening versus macrophysical changes such as cloud cover and LWP, which will be discussed further in this section.

### 4.1   Can a regional model produce a realistic representation of stratocumulus cloud when compared to a diverse
range of observations?

We have shown that the UM regional model with the sub-grid cloud scheme reproduced many important physical observations for the control case. The shape and magnitude of the observed diurnal cycle of domain mean LWP was captured to within $\sim$10 $gm^{-2}$ for the control run for the nighttime, but with an overestimate for the daytime of up to 30 $gm^{-2}$. The shapes and sizes of the observed bands of clouds were (qualitatively) reproduced and the model simulated open–cell–like regions of

low areal cloud cover to the NE of the domain and cloudy bands in the SW in between (Fig. 6). The daytime cloud fraction ($f_c$) frequency distribution, especially for the larger cloud fraction values ($f_c > \sim 0.7$) was reproduced to within a $f_c$ of 0.04, as was the domain mean nighttime $f_c$ at a single time (to within a $f_c$ of 0.02). Frequency distributions of shortwave and longwave TOA fluxes were close to those from observations; the means were underestimated by 15% and 0.4%, respectively. The higher of the two observed modes of the frequency distribution of droplet number concentrations was reproduced in the

control case with the mode value agreeing to within $\sim$20%. Also, the good comparison (to within $\sim$15%) of the width of the droplet concentration distribution indicates a good representation of model updraft velocities and the physics of the aerosol





activation process, although we also acknowledge that the aerosol mass and number concentrations in the control model run were uniformly scaled to approximately match the observed droplet concentrations.

Thus, there is good evidence that the model is correctly capturing the physical processes that are of first–order importance for producing a realistic stratocumulus deck. However, there are some model deficiencies that were highlighted in the comparison
to the observations that we now discuss.

Section 3.2.2 detailed how the daytime control run had a tendency to overestimate the LWP, particularly at the times of the lowest observed LWP. Examination of the PDF of LWP from a time when this problem occurred (14:12 LST on 13th Nov, Fig. 8a) revealed a lack of LWP values between around 15 and 70 $gm^{-2}$ and too many of the higher LWP ($>\sim125\ gm^{-2}$) values. A similar problem occurs at night (Fig. 8b) with a lack of LWP values between 150 and 250 $gm^{-2}$ and was also
evident from the comparisons to the longer term (day and night combined) single location ship observations (Fig. 9). This overestimation of LWP may be related to the lack of a lower mode of modelled $N_d$ values, which is evident from the $N_d$ PDF (Fig. 3). It is likely that the presence of the low $N_d$ mode in reality caused LWP removal through precipitation, which would lead to a reduction in the higher LWP values. The latter occurred in the lower aerosol runs in the model (CASIM-Ndvar-0.1 and CASIM-Ndvar-0.025), which were closer to the observed frequencies for the highest LWP values, and so if the lower observed
$N_d$ mode was present in the control model case (along with the higher mode) then we would expect the match to observations to improve. The dual mode of $N_d$ that was observed in reality could have been the result of a spatial gradient in $N_d$ (Fig. 2c), which is not captured by the model since we employ a uniform aerosol field. The introduction of a spatial aerosol gradient or the use of a realistic aerosol model that simulates the aerosol sources, transport and chemical transformation may rectify this problem.

BA12 found a similar daytime overestimate of LWP, but were only considering the near–coastal region where the ship was located. The reason for this was attributed to the sub-grid cloud scheme, which created too much cloud when supplied with the observed thermodynamic profiles. Since we use the same cloud fraction approach as BA12, albeit linked to a different microphysics scheme, this may also be an issue in this work. However, we note that the run with the old microphysics scheme (Old-mphys), which will be similar to the runs in BA12 since the same microphysics and cloud schemes are used, shows
a domain-mean LWP value that is quite similar to that observed at the time of the daytime minima. This suggests that the overestimate in the near–coastal region that was observed in BA12 is not having a large impact on the overall domain mean. In addition, Fig 4 clearly shows that the aerosol concentration has an impact on the LWP at this time with lower aerosol concentrations reducing the LWP significantly.

Another issue with the model was that the cloud top heights were too low compared to shipborne radar observations (Fig. 13).
This is consistent with BA12 where the UM model boundary layer height was found to be too low for this case through comparisons to radiosondes released from the ship, and is also consistent with Abel et al. (2010) where the UM boundary layer height was on average $\sim$200 m too low during the VOCALS field campaign period for NWP configuration runs at 0.15° resoluion. BA12 found that an improved treatment of rain microphysics increased the height of the boundary layer in their 1 km resolution nest through the suppression of precipitation. The improvements led to increased instances of coupled
boundary layers as diagnosed by the boundary layer scheme. In our case, though, the boundary layer is too low even in the





very high aerosol case (CASIM-Ndvar-10) when precipitation has already been completely suppressed indicating that this is not the cause of the low boundary layer in our runs. The most likely source of this discrepancy is the meteorology of the global driving model, which imposes an initial capping inversion on the 1 km nest that is too low. This could be reinforced through the lateral boundary conditions, which, even if the physics of the inner nest wanted to, might prevent it from growing

its boundary layer through entrainment. Another possibility is that both the global model and the inner nest do not produce enough entrainment and that it is this that leads to boundary layers that are too low. This is consistent with the results that are discussed in Section 4.2 whereby the modelled LWP in the high resolution nest does not decrease at very high aerosol concentrations.

The fact that the modelled boundary layer height was too low is also consistent with the overestimate of the LW TOA upwards

fluxes during both the daytime and the nighttime (Figs. 11 and 12) since it would correspond to cloud top temperatures that were too large. It is possible that the LW TOA overestimate is also indicative of a cloud fraction that is too low meaning that more surface points are contributing. However, this seems less likely given that the cloud fraction generally agreed well with the observations during both the daytime and the nighttime, although we note that the nighttime comparison of cloud fraction is limited to a single time.

The results presented in this paper suggest that the UM regional model with a relatively coarse horizontal and vertical resolution (1 km and ∼100 m at the top of the boundary layer, respectively) can reproduce most of the observed cloud characteristics for the case presented and thus is producing a realistic representation of stratocumulus clouds. However, we also emphasize that a good agreement was only found when employing a sub-grid cloud scheme to represent sub-grid variability in relative humidity. This is discussed further in Section 4.4.

**4.2   How do the modelled clouds respond to aerosol?**

Fig. 14 summarizes the response of various domain and time mean cloud properties to cloud droplet concentration ($N_d$) with the $N_d$ changes being driven by aerosol changes. The mean $SW_{upTOA}$ increases monotonically with $N_d$ with the value for CASIM-Ndvar-10 being more than twice that in CASIM-Ndvar-0.025. The other cloud properties show a non–monotonic increase across all $N_d$ values. Both the grid-box mean LWP and the in-cloud LWP ($LWP_{ic}$) increase at lower $N_d$ values

with, respectively, 84% and 52% increases between the- CASIM-Ndvar-0.025 and CASIM-Ndvar cases when the aerosol was increased by a factor of 40. In contrast, there was very little increase in LWP between the CASIM-Ndvar and CASIM-Ndvar-10 cases when the aerosol was increased by a factor of 10.

This variation in response is due to the influence of aerosol on the suppression of rain. As shown in Fig. 14, at low $N_d$ rain production (RWP) is relatively large, so increasing aerosol leads to a reduction in rain production and an increase in LWP.

When the mean $N_d$ reaches around $200 \ cm^{-3}$ (CASIM-Ndvar case) rain production has almost completely shut off, so any further increases in $N_d$ have no impact on LWP. Such behaviour is consistent with high resolution LES studies (e.g. A04) The simulations presented here show that a coarser resolution NWP model can simulate this behaviour. We also note that in some of the situations examined in A04, when $N_d$ was increased from very low to high levels they simulated a similar proportional increase in LWP to that which we observe between the CASIM-Ndvar-0.025 and CASIM-Ndvar runs.





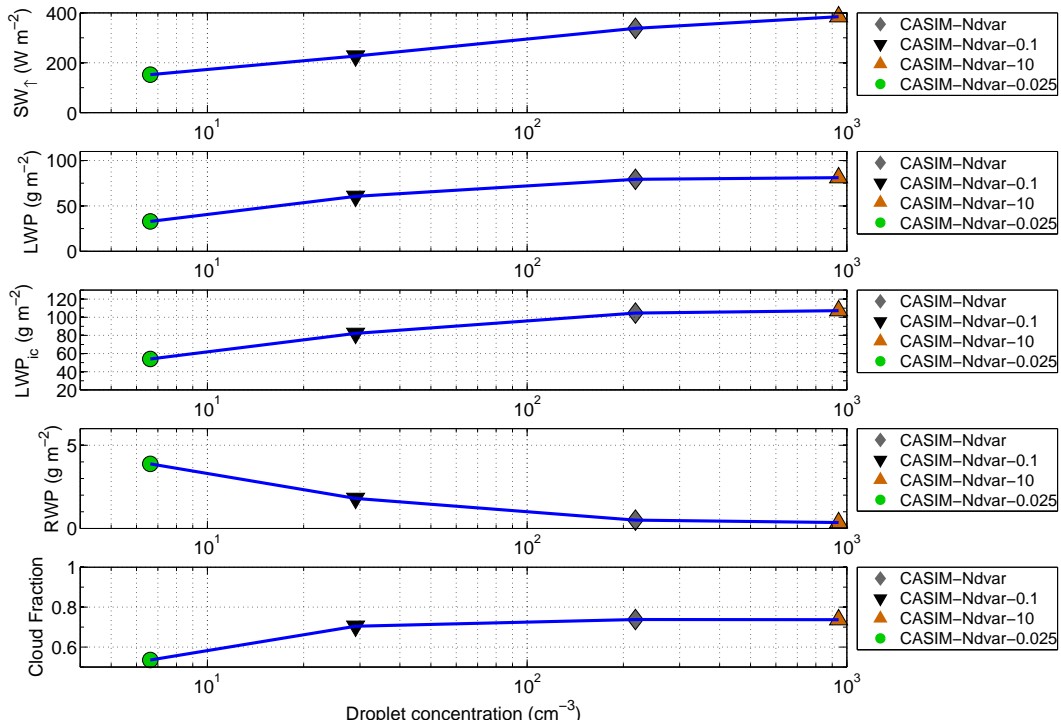

**Figure 14.** Summary plots of domain and time mean quantities for the different model runs. The time average is weighted by the incoming SW TOA flux timeseries to give extra weighting to times when the cloud properties will be contributing to the SW flux, i.e. mainly in the daytime. The x-axis shows droplet concentrations, which are calculated in the same way as for Fig 3. The top plot shows SW TOA flux, which is followed by LWP (grid-box mean), in-cloud LWP ($LWP_{ic}$), RWP and cloud fraction. Cloud fraction is calculated as for Fig. 10. Model data at 30 min intervals between the times of 06 UTC (01:12 LST) on 12th Nov and 0 UTC (19:12 LST) on 14th Nov are used, with the first six hours of model data being avoided due to spin-up.

The simulations in A04 also showed that once precipitation had been completely suppressed, LWP tended to decrease with further $N_d$ increases due to entrainment effects related to the ever smaller droplet sizes (Bretherton et al., 2007; Hill et al., 2009). However, we do not see such behaviour in our model in test runs (not shown) of the CASIM-Ndvar and CASIM-Ndvar-10 cases that include droplet sedimentation as a function of droplet size. This suggests that the entrainment parameterization within the boundary layer scheme (Lock et al., 2000) may need refinement to become more sensitive to the cloud droplet number concentration. Enhanced vertical resolution near the cloud top would also undoubtably improve the explicit representation of the entrainment interface layer, but such sensitivities are out of the scope of this study.





### 4.2.1 Cloud fraction response

Cloud fraction generally only increases between the CASIM-Ndvar-0.025 and CASIM-Ndvar-0.1 runs indicating that the change in LWP between those runs is due to an increase in both $f_c$ and $LWP_{ic}$. No major increase in $f_c$ occurs between the CASIM-Ndvar-0.1 and CASIM-Ndvar case, whereas there is a fairly large increase in $LWP_{ic}$ (27%) indicating cloud

thickening.

Thus, the cloud fraction exhibits a step change, which only occurs at very low $N_d$ values (mean values $<30\ cm^{-3}$ in our simulations). Such behavior makes it questionable whether aerosol cloud metrics where the response of cloud fraction to $N_d$ or aerosol is defined in terms of e.g. $dlog(f_c)/dlog(N_d)$ (e.g. Quaas et al., 2010, alebit this study used aerosol optical depth instead of $N_d$) is appropriate, at least across a wide range of $N_d$ values. The addition of aerosol processing to our model will

alter this behavior somewhat with the likelihood being that it will enhance the severity of the step change due to the positive feedback between CCN removal and precipitation rate ("runaway precipitation", B13). It may also shift the $N_d$ value at which it occurs to a higher value since small amounts of precipitation at higher $N_d$ are likely to be amplified by CCN removal. Therefore, we expect open–cell regions to be more likely to occur with aerosol processing operating.

### 4.3 What is the relative importance of macrophysical and cloud albedo changes for aerosol induced radiative effects?

Since the impact on $SW_{upTOA}$ of stratocumulus is perhaps the primary consideration it is useful to break down the aerosol-induced changes in this quantity into separate responses due to changes in $N_d$ (the cloud-albedo effect) and changes in macro-physical cloud properties such as $LWP_{ic}$ and $f_c$. A method for doing this using an analytical calculation of $SW_{upTOA}$ with the time and domain mean model cloud property values as inputs is described in Appendix C. Fig. 15 shows the results, where the runs shown in Fig. 14 have been compared to the runs with the next highest aerosol concentration. The closeness of the

red bars (changes in $SW_{upTOA}$ between runs from the on-line radiation code) to the totals from the values estimates using the analytical formulae suggest that the method works reasonably well.

The results indicate that the cloud albedo effect (i.e. the change due to $N_d$ alone with $LWP_{ic}$ and $f_c$ held constant) plays a large role in causing the changes in $SW_{upTOA}$ seen between the different runs, causing 44.5%, 69.9% and 94.7% of the total of the absolute changes (see Eqn. C7) for the three comparisons made in Fig 15. The contribution from $N_d$ changes is largest

for the comparison between the two highest $N_d$ runs since there is very little change in either $LWP_{ic}$, nor $f_c$ for those runs. As $N_d$ is reduced, the contribution from changes in $LWP_{ic}$ and $f_c$ increases, but the comparison between the two lowest $N_d$ runs indicates that the change due to $N_d$ is still the single largest factor. However, given that changes to $LWP_{ic}$ and $f_c$ can both be considered macrophysical cloud responses, and that there is some ambiguity in the definition of $LWP_{ic}$ and $f_c$ since it depends on the threshold chosen, their changes could be combined into one macrophysical response value. In that case the

percentage changes due to this combined macrophysical response would be 55.4%, compared to 44.6% for the cloud alebdo ($N_d$) effect, for the comparison between CASIM-Ndvar-0.025 and CASIM-Ndvar-0.1 suggesting that overall macrophysical responses are at least of equal importance when $N_d$ is low.





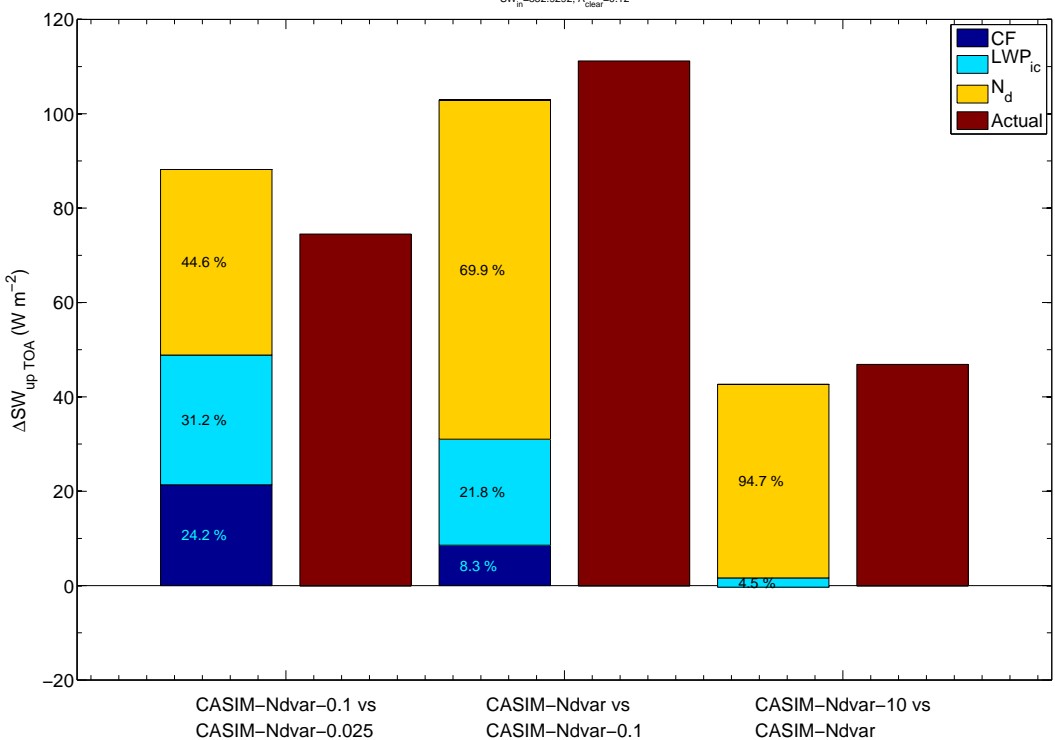

**Figure 15.** Estimated individual contributions to changes in the shortwave upwelling TOA radiation flux ($\Delta SW_{upTOA}$) due to changes in the domain and time mean (weighted as for Fig. 14) cloud properties (cloud fraction, $f_c$, in-cloud LWP, $LWP_{ic}$, and droplet number concentration, $N_d$, see the legend for the colour labelling) using an analytical formula for scene albedo. Changes in $SW_{upTOA}$ are estimated between a given run in Fig. 14 ($Run_i$) and the run with the next highest aerosol concentration ($Run_{i+1}$) by replacing one cloud property of $Run_i$ with that from $Run_{i+1}$; i.e. for the $f_c$ response, $\Delta SW_{f_c} = SW(f_{c_{i+1}}, LWP_{ic_i}, N_{d_i}) - SW(f_{c_i}, LWP_{ic_i}, N_{d_i})$ with the runs being compared indicated by the x-axis labels. The percentage that each cloud property contributes to the sum of the absolute changes is shown in the individual bars, e.g. the cloud fraction percentages are given by $100 \times \Delta SW_{f_c}/(|\Delta SW_{f_c}| + |\Delta SW_{LWP_{ic}}| + |\Delta SW_{N_d}|)$. The red bars show the actual change in SW from the model output.

## 4.4 What is the relative importance of the sub-grid cloud scheme?

For these simulations we introduced a sub-grid cloud scheme, which was shown in Fig. 4 to have a significant effect on cloud properties, even at the 1 km $\Delta x$ value used here. Domain LWP values with the cloud scheme switched off were lower than those of the CASIM-Ndvar-0.025 case (lowest aerosol run) for the first half of the simulation. They were higher for the second half, but were still lower than for the CASIM-Ndvar-0.1 case. Thus, the effect of the cloud scheme was comparable to that of large changes in the aerosol, which means that the inclusion of such a scheme within the UM is vital for the simulation of realistic marine stratocumulus clouds and therefore also for their response to aerosol. This may also be the case for other



models at this resolution. Given the importance of the cloud scheme it would be prudent to perform further investigation in future studies into the setting of the $RH_{crit}$ parameter (see Appendix A), or methods for deriving $RH_{crit}$ using, for example, sub-grid turbulent kinetic energy.

### 4.5  Model resolution considerations

The credible simulation of closed–cell stratocumulus using a horizontal resolution ($\Delta x$) of 1 km and a fairly coarse vertical resolution is consistent with previous results in the literature, for example BA12. Close matches to observations have also been reported using even coarser resolutions of 9 km (Yang et al., 2011) and 14 km (George et al., 2013) using the WRF-Chem model albeit with the representation of stratocumulus relying heavily on the boundary layer/stratocumulus parameterizations employed.

However, it is likely that a $\Delta x$ of 1 km or greater will not fully resolve some important effects, especially for cases that involve smaller scale eddies such as open–cell cases where aerosol removal and cold pool dynamics associated with narrow precipitating regions become more important. Model resolution also affects the spectra of vertical velocities that are represented, which has been shown to have an impact on the number of aerosol particles that are activated into droplets (e.g. Malavelle et al., 2014) and also has dynamical implications.

Despite this, we note that LES simulations looking at such aspects have been successfully performed, with little sensitivity to using a higher resolution, when using $\Delta x$ values of 200 m (Feingold et al., 2015), which is reasonably close to the 1 km resolution used here. Indeed, we do observe some features that resemble open–cells in our simulation (see Section 3.2.3) indicating that they start to be resolved using $\Delta x$=1 km. A factor to bear in mind here is that the different dynamical solvers used by different models may lead to a varying "effective resolution" for a given actual grid size. More work is required in

order to clarify these issues. In the future we will investigate the model performance for an open–cell or POC case at a range of model resolutions (with aerosol processing included).

As discussed above, our fairy coarse vertical resolution is not adequate to explicitly simulate the entrainment process; previous studies have shown (Stevens and Bretherton, 1999) that a vertical resolution of around 5 m is needed to do this. Therefore, our simulations will rely upon the boundary layer scheme to represent this process. Further work is needed to

investigate how well this scheme works and how it interacts with aerosol loading and droplet sizes, whether it is feasible to run over large domains with a very high resolution vertical grid near cloud top, and if so, how it affects model performance. Issues regarding the ratio of horizontal and vertical grid sizes are also likely to require further attention.

### 5  Conclusions

Stratocumulus clouds are very important for the earth's radiative budget. Aerosols form an integral part of the stratocumulus

system and aerosol perturbations can significantly alter the radiative properties of these clouds. Thus the realistic simulation of stratocumulus and its interaction with aerosol is vital for weather and climate predictions. In this paper we have addressed the question of whether the UM regional model with a new microphysics scheme and newly coupled sub–grid cloud scheme





can produce such a realistic representation of closed–cell stratocumulus and its response to aerosol when employing relatively coarse horizontal and vertical resolutions (1 km and 100 m at the inversion, respectively).

We compared UM runs with the recent CASIM microphysics scheme, along with a newly implemented sub-grid cloud scheme against a range of observational metrics. The run with control aerosol concentrations captured the shape of the domain mean LWP diurnal cycle as observed by satellite microwave instruments and agreed quantitatively for most of the diurnal cycle being within around 10 $gm^{-2}$ (∼10%) at the nighttime maxima. However, an overestimate of 20–30 $gm^{-2}$ (∼50–75%) was observed for the time of the daytime minima, which we suggest is due to a lack of spatial heterogeneity in the imposed model aerosol field leading to too many cloud droplets offshore and causing a lack of precipitation and hence a lack of LWP removal in those regions. This issue was highlighted by PDFs of model cloud droplet concentrations ($N_d$) that showed a high $N_d$ mode in agreement with the observations (for the near–shore region), but a lack of the low $N_d$ mode that was observed in the offshore region.

Daytime cloud fraction distributions from the model matched those from the GOES-10 satellite very closely for the control and high aerosol cases, especially for cloud fraction values >∼0.5. For these runs, the domain mean cloud fraction during the night also matched that observed to within 3 %. PDFs of shortwave TOA fluxes from the control model were very close to those from CERES, except for a slight underestimate in the frequency of values between around 500 and 700 $Wm^{-2}$, which analysis suggested was due to near–shore aerosol concentrations that were too low for the particular times of the CERES overpasses.

Radar observations showed that the modelled and observed cloud depths were quite similar (agreement within ∼18 %), as were the maximum reflectivities attained (-1.9 dBZ in the model vs -5.2 dBZ in the observations), the latter suggesting that the maximum droplet/raindrop embryo sizes reached in the model were close to those in reality. Model reflectivities also increased with height in a similar manner to the observations in the lower portion of the clouds. However, there were too many low reflectivity values in the model and the height of the modelled boundary layer was too low. The latter is a problem that was likely inherited from the driving global model, which indicates meteorology problems and/or a lack of entrainment in the global model and possibly in the nested model too. The maximum reflectivity was observed to decrease with height towards the cloud top, which was not the case for the models. Possible causes for this model bias may be a lack of entrainment, issues related to the incorrect model boundary layer height, or microphysical issues regarding rain formation.

Our model simulated a monotonic increase in the domain and time mean shortwave TOA flux ($SW_{upTOA}$) with increasing aerosol concentrations, with values more than doubling between the lowest and highest aerosol case, reinforcing that aerosol impacts are likely to drastically change the radiative properties of stratocumulus clouds. When aerosol was increased between the two runs with the lowest aerosol concentrations the aerosol change caused both a large cloud fraction increase and a large in–cloud LWP (cloudy sky ony) increase, which we showed was responsible for around half of the observed increase in $SW_{upTOA}$. Thus, the cloud macrophysical response was very important for this aerosol range. The rest of the $SW_{upTOA}$ increase was due to the increase in cloud albedo (i.e. droplet number concentration, $N_d$, alone). Further increases in aerosol caused only very small cloud fraction increases, suggesting that the cloud fraction response occurs over a fairly narrow range of aerosol concentrations, and that traditional ACI metrics may not be entirely appropriate for characterizing this.





The in–cloud LWP ($LWP_{ic}$) response also diminished with increasing aerosol concentration, such that the $N_d$ increase (i.e. the cloud albedo effect) became the dominant effect in terms of causing the observed changes in $SW_{upTOA}$. In fact the cloud albedo effect was strong throughout the entire range of aerosol concentrations that were tested, showing that, whilst this process is arguably more simple than the cloud macrophysical response, it is still important to simulate correctly. Furthermore,

for stratocumulus clouds it likely dominates over a wider range of aerosol concentrations than macrophysical responses. Nevertheless, the large macrophysical response of a reduction of cloud fraction and LWP at low aerosol concentrations may be very important in more pristine regions such as offshore from the SE Pacific coastal (VOCALS campaign) region, or in the Southern Ocean, for example. Large sensitivities of cloud radiative effects to aerosol may be expected within this low aerosol regime due to the cloud macrophysical response.

This study demonstrates the importance of using a sub–grid cloud scheme within the UM model for stratocumulus, even at 1 km horizontal resolution. Without the cloud scheme mean LWP values were up to around 50% too small, which is a difference that is comparable to that between the lowest and highest aerosol runs (representing an increase in aerosol by a factor of 400) during the first half of the simulation. It may also be the case that a cloud scheme needs to be considered for other aerosol-cloud interacting regional models in order to simulate realistic stratocumulus macrophysical properties.

The use of lower resolution paves the way for larger area, longer timescale simulations than has been previously possible with very high resolution LES models, or may even allow global simulations. Domain size could be important to allow the representation of large meteorological features and dynamical feedbacks between large area features such as between open–cell and non–open–cell regions, as well as for examining the wider scale dynamical impact of cloud–aerosol interactions. Thus, the realistic meteorology of our model represents an important advantage over LES models, which generally employ idealised

set-ups that do not allow spatially inhomogeneous meteorological forcing. Since the likelihood is that meteorology has an important influence on stratocumulus and its albedo, and that correlations exist between aerosol and meteorology, then the correct representation of meteorology is likely vital when considering cloud–aerosol interactions in a realistic setting. Once the planned coupling with a detailed aerosol emission, transport and chemistry model has taken place correlations between meteorology and aerosol, and their impact on cloud properties, can be examined.

It is envisioned that the model described here will facilitate the development of sub–grid parameterizations for the aerosol–cloud interactions processes described above for the global model. The use of nested high resolution model embedded within an operational model framework, such as is employed here, will allow straightforward testing of the parameterizations against observations since the global and nested models share the same meteorology.

## 6 Data availability

Raw model data is kept on tape archive available through the JASMIN (http://www.jasmin.ac.uk/) service. Please see http://www.ceda.ac.uk/blog/access-to-the-met-office-mass-archive-on-jasmin-goes-live/ for details on how to arrange access to Met Office data via JASMIN.



**Table 1.** The model vertical grid spacing (dz) as a function of height (z) for the boundary layer.

| Model level | z (m) | dz (m) |
|---|---|---|
| 1 | 2.50 | 2.50 |
| 2 | 13.33 | 10.83 |
| 3 | 33.33 | 20.00 |
| 4 | 60.00 | 26.67 |
| 5 | 93.33 | 33.33 |
| 6 | 133.33 | 40.00 |
| 7 | 180.00 | 46.67 |
| 8 | 233.33 | 53.33 |
| 9 | 293.33 | 60.00 |
| 10 | 360.00 | 66.67 |
| 11 | 433.33 | 73.33 |
| 12 | 513.33 | 80.00 |
| 13 | 600.00 | 86.67 |
| 14 | 693.33 | 93.33 |
| 15 | 793.33 | 100.00 |
| 16 | 900.00 | 106.67 |
| 17 | 1013.33 | 113.33 |
| 18 | 1133.33 | 120.00 |
| 19 | 1260.00 | 126.67 |
| 20 | 1393.33 | 133.33 |
| 21 | 1533.33 | 140.00 |

**Table 2.** CASIM microphysics scheme parameterization summary.

| Parameterization | Reference |
|---|---|
| Aerosol activation | Abdul-Razzak and Ghan (2000) |
| Autoconversion of droplets to rain | Khairoutdinov and Kogan (2000) |
| Accretion of droplets by rain | Khairoutdinov and Kogan (2000) |
| Rain self–collection | Beheng (1994) |



**Table 3.** The microphysical parameters used in the simulations for the equations described in Shipway and Hill (2012). $\rho_w$ is the density of water.

|  | Cloud | Rain |
|---|---|---|
| **Moment description parameters** | | |
| $p_1$ | 0 | 0 |
| $p_2$ | 3 | 3 |
| **Size spectra paramters** | | |
| $\mu$ | 0 | 2.5 |
| **Mass–diameter parameters** | | |
| $c_x$ | $\pi\rho_w/6$ | $\pi\rho_w/6$ |
| $d_x$ | 3 | 3 |
| **Fall–speed parameters** | | |
| $a_x$ | $3 \times 10^7$ | 130 |
| $b_x$ | 2 | 0.5 |
| $f_x$ | 0 | 0 |
| $g_x$ | 0.5 | 0.5 |

**Table 4.** UM model runs. "Standard $RH_{crit}$" refers to the standard profile (listed in Table A1) of the $RH_{crit}$ parameter that is used within the sub-grid cloud parameterization (see Appendix A)

| Model label | Description |
|---|---|
| Old-mphys | Old ("3D") microphysics, with standard $RH_{crit}$ |
| CASIM-Ndvar (CONTROL) | CASIM microphysics, variable $N_d$, standard $RH_{crit}$ |
| CASIM-Ndvar-0.025 | CASIM-Ndvar with aerosol x 0.025 |
| CASIM-Ndvar-0.1 | CASIM-Ndvar with aerosol x 0.1 |
| CASIM-Ndvar-10 | CASIM-Ndvar with aerosol x 10 |
| CASIM-Ndvar-RHcrit0.999 | CASIM-Ndvar with cloud scheme OFF |

**Table 5.** Details on the observations used in this study.

| Instrument | Measurements used | Product | Product resolution | Native pixel resolution | Estimated error | Reference |
|---|---|---|---|---|---|---|
| Microwave radiometers (AMSR-E, SSMI/SSMIS, TMI, Windsat) | LWP (Liquid Water Path) | Daily gridded | $0.25^o$ | $\sim 37 \times 28$ km for lowest resolution instrument | 2–3 $g\,m^{-2}$ (Lebsock and Su, 2014) | www.remss.com |
| GOES-10 | $N_d$ (droplet concentration), LWP | VOCALS region dataset | 4 km | 4 km | 20% mean low bias, r=0.91, RMSE = 36 $cm^{-3}$ (Painemal et al., 2012) | Wood et al. (2011b) |
| GOES-10 | LWP | VOCALS region dataset | 4 km | 4 km | 14.7% mean high bias, r=0.84, RMSE=26.9 $g\,m^{-2}$ (Painemal et al., 2012) | Wood et al. (2011b) |
| CERES | LW and SW TOA fluxes | SSF Level-2 snapshots | 20 km | 20 km | 5% for SW, 3% for LW (Loeb et al., 2007) | http://ceres.larc.nasa.gov/products.php?product=SSF-Level2 |
| Microwave radiometer (shipborne; stationed at 20° S, 75° W) | LWP | Vertical profiles every 10 minutes | N/A | N/A | Unknown | Zuidema et al. (2005); de Szoeke et al. (2012) |
| W–band (94 GHz) radar (shipborne; stationed at 20° S, 75° W) | Radar reflectivity (dBZ) | Vertical profiles every 10 minutes | N/A | N/A | Unknown | Moran et al. (2011); de Szoeke et al. (2012); Fairall et al. (2014) |




## Appendix A: Sub-grid cloud scheme

The sub-grid cloud scheme is based on the scheme described in Smith (1990, hereafter S90). We present only an outline for brevity and refer the reader to S90 for details. The basic assumption is that sub-grid fluctuations in liquid temperature ($T_L$) and/or total water mass mixing ratio ($q_T$) about the grid-box average can give rise to a sub-grid contribution to the total

condensed liquid water mass mixing ratio ($q_{Ltot}$) and cloud fraction ($f_c$) within a model grid-box. $T_L$ and $q_T$ are defined as :-

$$T_L = T - \frac{L_c}{c_p} q_L, \tag{A1}$$

$$q_T = q_v + q_L \tag{A2}$$

, where (for a given sub-grid element) $q_v$ and $q_L$ are the vapour and liquid mass mixing ratios and $T$ is the temperature; $L_c$ is the latent heat of condensation; and $c_p$ is the specific heat capacity of air at constant pressure. Note, that $T_L$ and $q_T$ are

conserved variables during the condensation process. Each given element of the sub-grid distribution has a liquid water content given by

$$q_L = \begin{cases} 0, & s \le -Q_c \\ Q_c + s, & s > Q_c \end{cases} \tag{A3}$$

, where $Q_c$ is the contribution to the liquid water mixing ratio from the grid-box mean quantities and $s$ represents the contributions due to perturbations about the grid-box mean state. The condition of $q_L = 0$ for $s \le -Q_c$ prevents $q_L < 0$. $Q_c$ is

given by :-

$$\begin{aligned} Q_c &= a_L(\overline{q_T} - q_{sat}(\overline{T_L}, \overline{p})) \\ &= a_L q_{sat}(\overline{T_L}, \overline{p})(\overline{RH_{tot}} - 1), \end{aligned} \tag{A4}$$

Here $q_{sat}$ refers to the saturation mixing ratio for liquid, $a_L = \frac{1}{1 + L_c \alpha_L / c_p}$ and $\alpha_L = \frac{\partial q_{sat}}{\partial T}\big|_{T=T_L} \approx \frac{\epsilon L_c q_{sat}(T_L, P)}{RT_L^2}$, $\overline{RH_{tot}} = \overline{q_T} / q_{sat}(\overline{T_L}, \overline{p})$ and overbars denote grid-box mean quantities. Note that $Q_c$ is negative for sub-saturated conditions (in terms of $q_T$, i.e. $\overline{RH_{tot}} < 1$).

$s$ is given by (Eqn. 2.20 of S90):-

$$s = a_L(q_T' - \alpha_L T_L'), \tag{A5}$$

,where $q_T'$ and $T_L'$ are the perturbations of the total water content and liquid temperature of the sub-grid element from the grid-box mean. The sub-grid distribution of $s$ is specified as an assumed probability density function (PDF) denoted as





$G(s)$. Note, that this formulation means that the PDFs of $q_T$ and $T_L$ do not need to be specified directly, just the PDF of $s$. For simplicity, this is assumed to be a triangular shaped function that is symmetric about $s = 0$, where $s = 0$ represents no perturbation from the mean and therefore is the mean state. The half-width is specified as $b_s$. Figure A1 depicts $G(s)$. Normalisation of $G(s)$, i.e. $\int_{-b_s}^{b_s} G(s)ds = 1$, dictates that $G(0) = 1/b_s$, from which it follows that :-

$$5 \quad G(s) = \begin{cases} 0, & s \leq -b_s \\ \frac{b_s+s}{b_s^2}, & -b_s \leq s < 0 \\ \frac{b_s-s}{b_s^2}, & 0 \leq s < b_s \\ 0, & s > b_s \end{cases} \tag{A6}$$

This sub-grid distribution of G(s) is assumed to exist in all grid-boxes. However, cloud will only be formed for the part of the distribution where $Q_c + s > 0$ (Eqn. A3). Thus the cloudy part of the sub-grid distribution will lie between $s = -Q_C$ and $s = b_s$ such that the cloud fraction ($f_c$) is given by :-

$$f_c = \int_{-Q_c}^{b_s} G(s)ds \tag{A7}$$

10      Therefore, for there to be cloud requires that $Q_c > -b_s$. Fig. A1 depicts the transition from a non-cloudy to a cloudy state as $RH_{tot}$ increases. In a highly sub-saturated mean state (in terms of total water, i.e. $\overline{RH_{tot}} << 1$), $Q_c$ will be a large negative number such that the start of the integral above will be a large positive number and will lie outside the maximum value in the $s$ distribution ($b_s$). As $Q_c$ increases, $-Q_c$ will decrease until it lies within the $s$ distribution. The cloud fraction will then be evaluated for the right-hand portion of the $G(s)$ distribution (the shaded part depicted in Fig.A1). $\overline{RH_{tot}}$ at the point at which 15   $-Q_c = b_s$ (i.e. where cloud begins to form) is denoted as the critical total relative humidity $RH_{crit}$. Thus from Eqn. A4 and the definition of $\overline{RH_{tot}}$ we can write :-

$$b_s = -a_L(\overline{q_T} - q_{sat}(\overline{T_L}, \overline{p}))$$
$$= a_L q_{sat}(\overline{T_L}, \overline{p})(1 - \overline{RH_{crit}}) \tag{A8}$$

, so that $b_s$ can also be specified in terms of $\overline{RH_{crit}}$, $\overline{T_L}$ and $\overline{p}$. We use the same $RH_{crit}$ values as used for the UM operational model, which are given in Table A1.



The solution for $f_c$ is (by combining Eqn. A6 and Eqn. A7) :-

$$f_c = \begin{cases} 0, & Q_N \leq -1 \\ \frac{1}{2}(1+Q_N)^2, & -1 < Q_N \leq 0 \\ 1 - \frac{1}{2}(1-Q_N)^2, & 0 < Q_N \leq 1 \\ 1, & 1 \leq Q_N \end{cases} \tag{A9}$$

, where we define $Q_N = Q_c/b_s$. $Q_N$ can be written as a function of $\overline{RH_{tot}}$ and $\overline{RH_{crit}}$ as follows (using Eqn. A4 and Eqn. A8:-

$$Q_N = \frac{Q_c}{b_s} = \frac{a_L(\overline{q_T} - q_{sat}(\overline{T_L}, \overline{p}))}{a_L q_{sat}(\overline{T_L}, \overline{p})(1 - \overline{RH_{crit}})} = \frac{\overline{RH_{tot}} - 1}{1 - \overline{RH_{crit}}} \tag{A10}$$

, which are both constant during the cloud formation process, and so $f_c$ can be calculated directly from the apriori grid-box mean values.

The mean liquid water mass mixing ratio is calculated as follows:-

$$\overline{q_L} = \int_{-Q_c}^{b_s} (Q_c + s)G(s)ds \tag{A11}$$

10    The solution to this is :-

$$\frac{\overline{q_L}}{b_s} = \begin{cases} 0, & Q_N \leq -1 \\ \frac{1}{6}(1+Q_N)^3, & -1 < Q_N \leq 0 \\ Q_N + \frac{1}{6}(1-Q_N)^3, & 0 < Q_N \leq 1 \\ Q_N, & 1 \leq Q_N \end{cases} \tag{A12}$$

Thus, the solution for $\overline{q_L}$ contains the $b_s$ term, which is calculated from $Q_c$ via Eqns. A4 and A8. Equation A4, which defines $Q_c$, is based upon a first order Taylor expansion approximation of $q_{sat}(T)$ as a function of $q_{sat}(T_L)$, since the final value of $T$ is unknown apriori as it is a function of $q_L$, which has yet to be determined. Inaccuracy due to this approximation

15  can be improved by an iterative procedure, which is described fully in Wilson (2011).

Advection operates upon the grid-box mean mixing ratio and number concentrations of liquid water (denoted $\overline{q_L}$ and $\overline{n_L}$). However, before the calculation of the microphysical process rates $\overline{q_L}$ and $\overline{n_L}$ are divided by $f_c$ so that the mean values over only the cloudy part of the grid-box are used. Once the process rates have been calculated, the ones that change $\overline{q_L}$ and $\overline{n_L}$ are multiplied by $f_c$. This ensures that any non-linearity in the microphysical processes is captured, i.e. cases where




$f_c \times P(\frac{\overline{q_c}}{f_c}, \frac{\overline{n_c}}{f_c}) \neq P(\overline{q_c}, \overline{n_c})$, where $P(q_c, n_c)$ represents a microphysical process rate as a function of $q_c$ and $n_c$. The radiation scheme also takes into account $f_c$ for liquid clouds in a similar manner.

It is also possible to estimate a fraction over which precipitation is likely to present. For example Chosson et al. (2014) makes such an estimate based on the cloud fraction in the layers above. However, problems may arise due to separation of the precipitating regions from the cloudy regions above by horizontal advection. Thus, for this paper we have assumed a precipitation fraction of 1.0 for simplicity and will test the effect of a sub-grid precipitation fraction in subsequent work. Since we have concentrated on warm clouds here, no account of a sub-grid ice cloud fraction has been made.

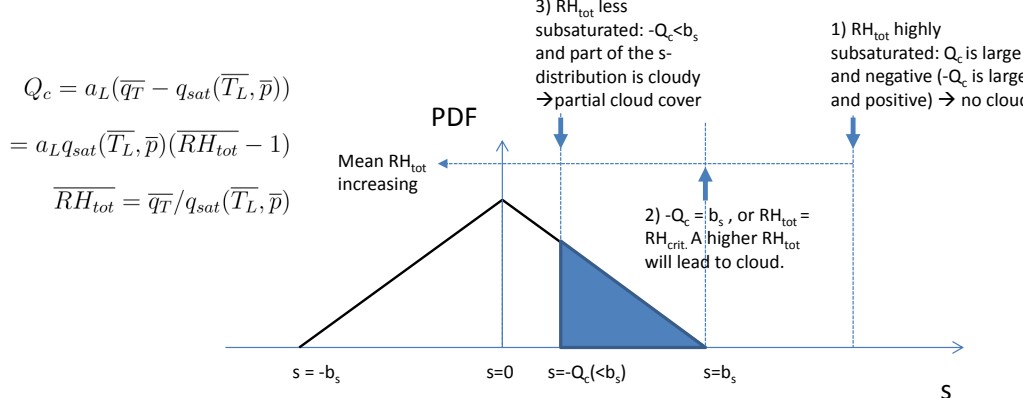

**Figure A1.** The assumed triangular shaped PDF, $G(s)$, to represent the sub-grid distribution of s (Eqn. A6), which is the liquid water mass mixing ratio associated with perturbations about the mean grid-box state (for which s=0). The half-width of the distribution is called $b_s$ and is a specified parameter. Cloud forms for the part of the distribution for which $q_L > 0$, which corresponds to $s > -Q_c$ (Eqn. A3), if $-Q_c < b_s$, with the cloud fraction given by $\int_{-Q_c}^{b_s} G(s)ds$ (shaded part of the figure; note, the shading applies to situation (3) as labelled in the figure, i.e. $-Q_c < b_s$). $Q_c$ is the amount of total water content above saturation for the grid-box mean state and thus increases as the grid-box mean total water relative humidity ($\overline{RH_{tot}}$) increases. For highly sub-saturated mean conditions (in terms of $\overline{RH_{tot}}$), $Q_c$ is large and negative such that $-Q_c > b_s$ and no part of the sub-grid distribution is cloudy (see (1) in the figure). With increasing $\overline{RH_{tot}}$ a critical relative humidity (denoted as $RH_{crit}$) is reached (2) where $-Q_c = b_s$. Further increases in $\overline{RH_{tot}}$ will lead to partial cloud coverage (e.g. (3)) with full cloud cover only being reached when $Q_c = b_s$, corresponding to $\overline{RH_{tot}} > 1$.





**Table A1.** The $RH_{crit}$ values used in the sub-grid cloud scheme.

| Model level | $\mathbf{RH_{crit}}$ |
|---|---|
| 1 | 0.96 |
| 2 | 0.94 |
| 3 | 0.92 |
| 4 | 0.9 |
| 5 | 0.89 |
| 6 | 0.88 |
| 7 | 0.87 |
| 8 | 0.86 |
| 9 | 0.85 |
| 10 | 0.84 |
| 11 | 0.84 |
| 12 | 0.83 |
| 13 | 0.82 |
| 14 | 0.81 |
| $\geq 15$ | 0.8 |



### Appendix B: Observation details

The observations used for the model evaluation in this paper are now described. See Table 5 for a summary.

There are several satellites that have microwave radiometer instruments onboard and which provide coverage of the study region. These instruments include AMSR-E (onboard Aqua), the SSMI/SSMIS instruments (onboard the f13, f15, f16 and f17 satellites), TMI (onboard TRMM) and Windsat (onboard Coriolis). These instruments report an overall average LWP for the cloudy and clear parts of a given region (i.e. no attempt is made to separate cloudy and clear pixels). We use the gridded daily data that are provided at $0.25 \times 0.25^{\circ}$ resolution from www.remss.com and are separated into daytime and nighttime overpasses. The native resolution for the 37 GHz LWP retrieval that is used for the LWP retrievals vary from instrument to instrument, with the lowest resolution being $\sim 37 \times 28$ km for the SSMI instruments. Snapshots within the study region are made by the different instruments at various times of the day, allowing a diurnal cycle to be built up from their combination. Further details on the specifics of the retrieval algorithms for each instrument can be found at http://www.remss.com/missions.

Microwave radiometers provide a fairly reliable estimate of cloud liquid water path (LWP), although some errors have been identified in the form of non-zero values being reported in clear-sky situations. However, examining AMSR-E data, (Lebsock and Su, 2014) reported that these errors were larger for situations with high column water vapour values; the mean clear-sky bias in the SE Pacific was only around 2–3 $gm^{-2}$.

Data from the GOES-10 geostationary satellite is also extensively used in our study. A special data set was created for the VOCALS field campaign that covers the location and period of the simulations performed in this study. The data were analyzed as in Minnis et al. (2008) using the methods of Minnis et al. (2011) as described in Wood et al. (2011b) and Allen et al. (2013). We mainly use the retrievals of cloud optical depth ($\tau$) and effective radius ($r_e$) that are provided at 4km spatial resolution every 30 minutes. From these two quantities we make an estimate of LWP following Wood and Hartmann (2006) :-

$$LWP = 5/9\rho_w \tau r_e, \tag{B1}$$

, where $\rho_w$ is the density of liquid water.

Estimates of cloud droplet concentrations are also made using the technique described in Grosvenor and Wood (2014, hereafter GW14). We only use the daytime data in this study since the retrieval of $r_e$ and $\tau$ when there is no daylight uses an experimental technique for which the reliability is not well proven. In addition, daytime retrievals where the solar zenith angle is larger than 65$^{\circ}$ are not used due to the likelihood of biases (GW14).

Painemal et al. (2012) performed a comparison of the GOES-10 $r_e$, $\tau$ and $N_d$ retrievals to aircraft and MODIS satellite observations for the VOCALS field campaign period. Since our simulations are performed during the time of VOCALS and very close to where the aircraft measurements took place, the GOES-10 errors reported should be representative of the errors that we can expect for our study. The GOES-10 $r_e$ was found to be well correlated (r=0.91) with the aircraft values, but biased high by 22%, which was a very similar positive bias to that reported for MODIS $r_e$ values. GOES-10 $\tau$ also correlated well with the aircraft observations (r=0.79), but had a much smaller mean bias of only 6%. LWP estimates using the above equation demonstrated a mean positive bias of 14.7% (attributable to the positive $r_e$ bias), an RMSE of 26.9 $gm^{-2}$ and r=0.84. The





mean bias for $N_d$ (calculated using the same method that we apply) was -20%, with r=0.91 and an RMSE of 36 $cm^{-3}$; this fairly low bias was achieved despite the strong dependence of $N_d$ on $r_e$ ($N_d \propto r_e^{-2.5}$) and the relatively high $r_e$ bias, due to compensating biases in some of the other factors used to estimate $N_d$, as described in Painemal and Zuidema (2011).

Longwave (LW) and shortwave (SW) top-of-the-atmosphere (TOA) radiative fluxes are obtained from the CERES instru-
ments (Wielicki et al., 1996) that are onboard both the Aqua and Terra satellites. We use the SSF Level-2 product (CERES_SSF_XTRK-MODIS_Edition4A, taken from http://ceres.larc.nasa.gov/products.php?product=SSF-Level2), which is provided at a nominal resolution of 20 km. These instruments provide snapshots of the radiative fluxes at local times (for the swath centre) of around 01:30 and 13:30 for the Aqua satellite and 10:30 and 22:30 for the Terra satellite, although we note that SW fluxes are only available during daylight hours. Also, CERES data is unavailable for the afternoon overpass on 13th November, 2008 (the
second day of our simulation period) for the Aqua instrument. Loeb et al. (2007) estimate an uncertainty in CERES fluxes of less than 5% for SW and less than 3% for LW for overcast, moderately thick, or thick low clouds over the ocean, which are the predominant cloud type in the region of our study.

The Ronald H. Brown research vessel was stationed near 20º S 75º W, which is near to the centre of our model domain. Data was gathered from the onboard instruments throughout the modelled period, which we use to help evaluate the model. A
microwave radiometer provided LWP estimates every 10 minutes (Zuidema et al., 2005; de Szoeke et al., 2012). The 94 GHz (W-band) cloud radar produced data at the same time frequency (Moran et al., 2011; de Szoeke et al., 2012; Fairall et al., 2014) and is used to evaluate model drizzle and large droplet properties.

**Appendix C: Shortwave radiation calculations**

The shortwave top-of-the-atmosphere (TOA) upwards radiative flux ($SW_{upTOA}$) is estimated from the domain and time mean
in-cloud LWP ($LWP_{ic}$), droplet concentration ($N_d$) and the cloud fraction ($f_c$) using analytical formulae. Firstly, the cloud optical depth ($\tau$) is estimated by assuming that the clouds are adiabatic (or some constant fraction of adiabatic) so that their liquid water increases linearly with height, and it is assumed that $N_d$ is constant throughout their depth. Observations suggest that both are valid assumptions for stratocumulus clouds (Albrecht et al., 1990; Zuidema et al., 2005; Painemal and Zuidema, 2011; Miles et al., 2000; Wood, 2005). With these assumptions $\tau$ can be related to $LWP_{ic}$ and $N_d$ by rearranging the formula
for $N_d$ given in GW14 (Eqn. A1 of that paper) and replacing the effective radius ($r_e$) using Eqn. B1 to give :-

$$\tau = \frac{(9/5)^{5/2} \pi k Q^3}{2\sqrt{10}\rho_w^2 c(T,P)^{1/2}} N_d^{1/3} LWP_{ic}^{5/6}$$

$$k = (r_v/r_e)^3, \tag{C1}$$

where $\tau$ is the cloud optical thickness, $r_e$ and $r_v$ are the cloud top effective and volume mean radius, respectively, $k$ is cube of the ratio of $r_v$ to $r_e$, $\rho_w$ is the density of water and Q is the scattering efficiency. Q has been shown to have a constant value very close to 2 for droplet radii that are much larger than the wavelength of light concerned (Bennartz, 2007). $c(T,P)$ is the
rate of increase of liquid water content ($q_L$) with height ($dq_L/dz$, with units $kgm^{-4}$ and is referred to as the "condensation





rate" in Bennartz (2007), or the "water content lapse rate" in Painemal and Zuidema (2011). See Ahmad et al. (2013) for a definition. The cloud top temperature as determined by GOES-10 is used for the temperature ($T$) in the calculation of $c(T, P)$, along with a constant pressure ($P$) of 850 hPa. GW14 show these two approximations are likely to cause very little error.

The cloud albedo ($A_c$) is then estimated using Eqn. 24.38 of Seinfeld and Pandis (2006), which is based on the two-stream approximation for a non-absorbing, horizontally homogeneous cloud :-

$$A_c = \frac{\tau}{\tau + 7.7} \tag{C2}$$

The shortwave upwards flux at cloud top ($SW_{upCT}$) for a given cloud fraction ($f_c$) can then be calculated as :-

$$SW_{upCT} = SW_{downCT}(f_c A_c + (1 - f_c)A_s) \tag{C3}$$

,where $SW_{downCT}$ is the SW downwelling flux at cloud top and $A_s$ is the surface albedo. $SW_{downCT}$ is approximated as the SW downwelling flux at the surface ($SW_{downSURF}$), which is estimated from the model data for clear columns. Thus, this estimate neglects any additional scattering or absorption between the typical cloud top heights and the surface. However, since the cloud top heights are low, this should not lead to a large error. $SW_{upTOA}$ can then be estimated as :-

$$SW_{upTOA} = T_r SW_{upCT} \tag{C4}$$

,where Tr is the transmission of the atmosphere, which is assumed constant and is estimated using :-

$$T_r = SW_{downSURF}/SW_{downTOA} \tag{C5}$$

Here we are assuming that the downwards transmission is equal to the upwards transmisison.

Using these formulae we calculated $SW_{upTOA}$ from the model domain and time mean cloud properties. Time means were weighted by $SW_{downTOA}$ in order to prioritise the daytime values. However, when weighted equally over all times the relationships between the different model runs (i.e. different $N_d$ values) were very similar, although the magnitudes of the averages was changed. We found that the analytical estimates were within 13% of the actual mean $SW_{upTOA}$ as calculated online by the model radiation code for the four model runs shown in Fig. 14. The analytical formulae were then used to estimate the individual effects on $SW_{upTOA}$ of the changes in $f_c$, $LWP_{ic}$ and $N_d$ between the different runs, with each run having different prescribed aerosol loadings. Changes in $SW_{upTOA}$ are estimated between a given run in Fig. 14 ($Run_i$) and the run with the next highest aerosol concentration ($Run_{i+1}$) by replacing one cloud property of $Run_i$ with that from $Run_{i+1}$; e.g. for the cloud fraction response,

$$\Delta SW_{f_c} = SW(f_{c_{i+1}}, LWP_{ic_i}, N_{d_i}) - SW(f_{c_i}, LWP_{ic_i}, N_{d_i}) \tag{C6}$$



The percentage that each cloud property contributes to the sum of the absolute changes ($P_x$) for a given $Run_i$ is also calculated (as shown as text in the individual bars in Fig. 15), e.g. for cloud fraction :-

$$P_{f_c} = \frac{100 \times \Delta SW_{f_c}}{|\Delta SW_{f_c}| + |\Delta SW_{LWP_{ic}}| + |\Delta SW_{N_d}|} \tag{C7}$$

*Author contributions.* D. P. Grosvenor, P. R. Field and A. A. Hill developed the concepts and ideas for the direction of the paper. B. S. Ship-
way wrote the CASIM microphysics code. D. P. Grosvenor, A. A. Hill and P. R. Field helped to further develop the CASIM microphysics
code and performed the coupling of the sub-grid cloud scheme parameterization. D. P. Grosvenor, P. R. Field and A. A. Hill helped set up
and complete the model runs. D. P. Grosvenor performed the model data analysis; complied and analysed the observational data sets; and
wrote the majority of the manuscript, along with input and comments by P. R. Field, A. A. Hill and B. S. Shipway.

*Competing interests.* The authors declare that they have no conflict of interest.

*Acknowledgements.* We acknowledge use of the MONSooN system, a collaborative facility supplied under the Joint Weather and Climate
Research Programme, a strategic partnership between the Met Office and the Natural Environment Research Council. The GOES-10 and the
R. H. Brown ship radiometer and radar data datasets were originally provided by NCAR/EOL under sponsorship of the National Science
Foundation NCAR/EOL and are available at http://data.eol.ucar.edu.NCAR/EOL. We thank Steve Abel and Ian Boutle for the assimilation
and provision of those radiometer and radar data sets and we thank Grant Allen for the assimilation and provision of the GOES-10 data set.
Peter Minnett is acknowledged for the use of his microwave radiometer onboard the R. H. Brown ship during the VOCALS field campaign.
AMSR-E, SSMI/SSMIS, Windsat and TMI data are produced by Remote Sensing Systems and sponsored by the NASA Earth Science
MEaSUREs DISCOVER Project and the NASA AMSR-E Science Team. Data are available at www.remss.com. Finally, we thank Stuart
Webster for his help setting up the nested suite of simulations.





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
