# Peer review of "The relative importance of macrophysical and cloud albedo changes for aerosol induced radiative effects in closed-cell stratocumulus: insight from the modelling of a case study"

_Atmospheric Chemistry and Physics, 2016_

## Referee Comment (RC1) · Anonymous Referee #1 · 23 Dec 2016

**1 General comments**

This paper uses a case-study simulation a 1 km resolution with the new CASIM microphysics scheme in the Met Office UM to address several questions of great interest to the cloud and aerosol communities, namely

1. how do stratocumulus clouds respond to aerosol?
2. what model resolution is required to simulate SCu realistically?
3. how important is the subgrid cloud cover scheme?

On the whole, the paper is well written, and the authors provide valuable answers to all of these questions, to the extent that a case study can answer them. Some questions arise in the manuscript that are not answered, but the authors promise to address them in follow-up work, which is appropriate. I have a few suggestions for minor clarifications (see below) and recommend publication once these have been addressed. I would also like to commend the authors for the level of detail provided in the appendices and for making the model output available.

**2 Specific comments**

1. The authors use the model to partition the cloud response to aerosols into "macrophysical" (cloud fraction and liquid water path) and "microphysical" (droplet size) responses. Being able to use the model to understand the various mechanisms at work is one of the major benefits of having a reliable model, so I feel this is an important part of the paper. Since the authors point out that their model works to their satisfaction only in closed-cell SCu (p. 13 l. 30f.), the title and abstract should reflect that fact. (The title and abstract should also reflect that the results are based on a model and reflect a case study.)

2. In Sec. 4.2.1, the distinction between LWP and $LWP_{ic}$ is made. This leads me to assume that LWP refers to gridbox-mean LWP throughout the manuscript. If this is not the case, the manuscript should be changed where appropriate.

3. The high model bias in LW fluxes is attributed to low bias in cloud altitude or cloud fraction (p. 20, l. 25). What about the cloud thickness? I realize that the effect of LWP on the LW flux probably saturates pretty quickly, but the modeled LWP peaks at pretty small values.

4. p. 15, footnote 1: more explanation is needed here; I assume "Poisson counting statistics” means that the uncertainty scales as $\sqrt{n}$, but that doesn't tell me whether the ranges quoted are $1\sigma$, 90%, 95%, etc. confidence intervals.

5. The authors are right to point out that the subgrid cloud scheme may play an important role even at fairly high resolution. However, one of the drawbacks of case studies is that it is difficult to tell which conclusions generalize (see my first specific comment above). Changing “demonstrates” to “suggests” on p. 32 l. 10 would make me feel more confident in the conclusion.

**3 Technical corrections**

The manuscript, while well written, would benefit from thorough proofreading. In addition, units are consistently italicized when they should be roman; I believe `copernicus.cls` provides the `\unit` command for this purpose.

---

## Referee Comment (RC2) · Anonymous Referee #2 · 4 Jan 2017

This manuscript presents a numerical study of aerosol effects on cloud, precipitation and radiation in stratocumulus region within the VOCAL experiment area. By running the Met Office UM model at 1-km grid spacing with the newly developed CASIM microphysics scheme and a sub-grid cloud scheme, the authors investigated the aerosol impacts under different background concentrations. It was found that a regional model running at 1-km grid spacing with a sub-grid cloud scheme driven by the realistic meteorological conditions is able to reproduce the observed properties of the stratocumulus and aerosol impacts stratocumulus through both macro- and microphysical responses when the loading is low and mostly through microphysical changes when the loading is high.

The manuscript is well written and organized. The questions explored here are of great interests of the community. The findings are interesting and novel, which makes it appropriate to be published in ACP. I would like the authors to consider the following points before the paper is accepted.

1. The title should reflect the fact that this is a numerical study.

2. Thorough validation of the model results against various observations is a key part of the paper to put more confidence in the simulated aerosol impacts. Therefore, environmental conditions simulated by the model should be validated as well. I suggest the authors to show the ship-borne sounding comparisons.

3. LES simulations driven by mesoscale models than incorporate large scale dynamic and thermodynamic structure are not that uncommon now. Chow et al. (2006) demonstrated the approach. Xue et al. (2014, 2016) showed that LES simulations of actual events reproduced observational features very well. Therefore, discussions about LES v.s. regional model in pages 3 and 32 should be adjusted.

4. What type of data from UM N512 was used to drive the 1-km simulations, analysis or forecast data (Section 2.1)?

5. The model data should apply the same way as how GOES-10 gets the 2D Nd field to calculate the 2D Nd field (Section 3.2.1).

6. Sections 3.2.3 and 3.2.4 should be simplified. Too much detail now.

7. Why don't you use all available satellite data to plot the LWP PDF in Fig. 8? The data sample in current Fig. 8 is very limited based on just one snapshot.

8. The 1X1 degree region in the model is too large to compare with the ship-borne radar CFAD. The W-band radar will not cover such a big area. A cloud regime matching technique can be used to choose the right area in the model results.

9. The RHcrit in the sub-grid cloud scheme should be a function of aerosol loading so

that the aerosol impact on sub-grid cloud can be addressed.

Some technical suggestions:

Page 1, lines 18 to 19 and later in Page 24: ... to within delta fc = 0.04 ... It is the fc difference not the fc itself.

Page 5, line 23: ... since the presence of some liquid ...

Page 7, line 2: ... investigate its impact. You shouldn't know whether it is important or not.

Page 8, caption of Figure 2: should the box be blue not white?

Page 17, line 20 and page 19 last line: ... for the entire of the ...

Page 20, line 5: ... similar to those observed by CERES.

Page 20, line 6: ... unlikely to be the result of cloud fraction ...

Page 20, line 7: ... distribution for this and higher ...

Page 26, line 6: ... and that leads to lower boundary layer.

Page 31, line 16: Please reword this sentence.

Page 31, line 30: , which was responsible ...

---

## Author Comment (AC1) · 1 Mar 2017

**Response to reviewer comments on "The relative importance of macrophysical and cloud albedo changes for aerosol induced radiative effects in stratocumulus" by Daniel P. Grosvenor et al.**

**Comments from anonymous Referee #1**

We sincerely thank the Referee for taking the time to review our paper and for providing constructive suggestions for improvement. We hope that we have fully addressed the Referee's comments – our responses are listed below in non-bold font.

*Specific comments*

**1. The authors use the model to partition the cloud response to aerosols into "macrophysical" (cloud fraction and liquid water path) and "microphysical" (droplet size) responses. Being able to use the model to understand the various mechanisms at work is one of the major benefits of having a reliable model, so I feel this is an important part of the paper. Since the authors point out that their model works to their satisfaction only in closed-cell SCu (p. 13 l. 30f.), the title and abstract should reflect that fact. (The title and abstract should also reflect that the results are based on a model and reflect a case study.)**

In response to this comment, and also one made by Referee #2, we have changed the title to :

 *"The relative importance of macrophysical and cloud albedo changes for aerosol induced radiative effects in closed-cell stratocumulus: insight from the modelling of a case study."*

, to include the fact that this was a modelling/numerical study of a case study and that we primarily are concerned with closed-cell stratocumulus.

The first sentence of the abstract is also modified from :-

"Aerosol-cloud interactions are explored using 1~km resolution simulations of SE Pacific stratocumulus clouds that include realistic meteorology along with newly implemented cloud microphysics and sub-grid cloud schemes."

to :-

"Aerosol-cloud interactions are explored using a 1~km resolution simulations of a case study of predominantly closed--cell SE Pacific stratocumulus clouds. The simulations include realistic meteorology along with newly implemented cloud microphysics and sub-grid cloud schemes."

**2. In Sec. 4.2.1, the distinction between LWP and LWPic is made. This leads me to assume that LWP refers to gridbox-mean LWP throughout the manuscript. If this is not the case, the manuscript should be changed where appropriate.**

Apologies, but we should have referred to this as the "domain mean LWP" rather than the grid-box mean since we were comparing the overall LWP change to that in the cloudy grid boxes only (rather than considering the sub-grid cloud fraction). This has been changed. The model LWP values in the rest of the paper are based upon the grid-box mean values for each 1km grid cell.

**3. The high model bias in LW fluxes is attributed to low bias in cloud altitude or cloud fraction (p. 20, l. 25). What about the cloud thickness? I realize that the effect of LWP on the LW flux probably saturates pretty quickly, but the modeled LWP peaks at pretty small values.**

Thanks for the suggestion, this has been added as a possibility :-

shifted to too high LW flux values, indicating either clouds that are too low in altitude, cloud fractions that are too low, or clouds that are too thin. However, we note that cloud thickness would only be relevant for very thin cloud regions ($< \sim 20 \ \mathrm{gm}^{-2}$) since the increase in LW flux with LWP saturates at low LWP values (Miller et al., 2015). As for the daytime results, there is a much

**4. p. 15, footnote 1: more explanation is needed here; I assume "Poisson counting statistics" means that the uncertainty scales as √n, but that doesn't tell me whether the ranges quoted are 1σ, 90%, 95%, etc. confidence intervals.**

We estimated the Poisson error bars for the frequencies of each histrogram bin based on ±√n. The ranges quoted were then the minimum and maximum (vice versa) differences in frequencies between the model and the observation. The minimum difference was calculated as the upper bound of the error bar for whichever was lowest out of the observation and the model minus the lower bound for whichever was highest. And vice versa for the maximum.

However, based on the request to simplify this section by Referee #2 we have removed the information about the degree of agreement using the Poisson statistics as we felt it was adding unnecessary detail.

**5. The authors are right to point out that the subgrid cloud scheme may play an important role even at fairly high resolution. However, one of the drawbacks of case studies is that it is difficult to tell which conclusions generalize (see my first specific comment above). Changing "demonstrates" to "suggests" on p. 32 l. 10 would make me feel more confident in the conclusion.**

We have changed this sentence to :-

"This study suggests that it may be necessary to employ a sub-grid cloud scheme within the UM model for stratocumulus, even at 1~km horizontal resolution. This finding may also apply to other models."

***Technical corrections***

**The manuscript, while well written, would benefit from thorough proofreading. In**

**addition, units are consistently italicized when they should be roman; I believe**

**copernicus.cls provides the \unit command for this purpose**

Thanks, we will ensure that the final manuscript is proofread and we have used the \unit command throughout.

**References**

Miller, N. B., Shupe, M. D., Cox, C. J., Walden, V. P., Turner, D. D., and Steffen, K.: Cloud Radiative Forcing at Summit, Greenland, Journal of Climate, 28, 6267–6280, doi:10.1175/jcli-d-15-0076.1, https://doi.org/10.1175%2Fjcli-d-15-0076.1, 2015.

---

## Author Comment (AC2) · 1 Mar 2017

**Response to reviewer comments on "The relative importance of macrophysical and cloud albedo changes for aerosol induced radiative effects in stratocumulus" by Daniel P. Grosvenor et al.**

**Comments from anonymous Referee #2**

We sincerely thank the Referee for taking the time to review our paper and for providing constructive suggestions for improvement. We hope that we have fully addressed the Referee's comments – our responses are listed below in non-bold font.

**1. The title should reflect the fact that this is a numerical study.**

In response to this comment, and also one made by Referee #1, we have changed the title to :

 *"The relative importance of macrophysical and cloud albedo changes for aerosol induced radiative effects in closed-cell stratocumulus: insight from the modelling of a case study."*

, to include the fact that this was a modelling/numerical study of a case study and that we primarily are concerned with closed-cell stratocumulus.

**2. Thorough validation of the model results against various observations is a key part**

**of the paper to put more confidence in the simulated aerosol impacts. Therefore, environmental**

**conditions simulated by the model should be validated as well. I suggest**

**the authors to show the ship-borne sounding comparisons.**

Thank you for the suggestion, which has led to a nice comparison that supports the arguments made concerning the CFAD results (i.e. the boundary layer being too low). We have now included a figure to show the comparison to the radiosondes :-

[revised manuscript text omitted]

**4. What type of data from UM N512 was used to drive the 1-km simulations, analysis or forecast data (Section 2.1)?**

The global model used to drive the 1km simulations was a 2-day forecast run, which was initialized using global UM analysis. Fields used to initialize the 1km nest and to force the boundaries included wind, temperature and condensed water fields. Section 2.1 has been modified as follows :-

equator for dx×dy) with 70 vertical levels below 80 km that are quadratically spaced giving more levels near the surface. This is run in forecast mode for two days (12-14th November) based on an initial field from the UM global operational analysis. The global run provides the initial conditions and forces the lateral boundaries for the wind, moisture, temperature and condensed water fields for a single 1 km resolution nest centred at 20º S, 76º W of size 600×600 km (Fig. 1). This places the domain near

**5. The model data should apply the same way as how GOES-10 gets the 2D Nd field to calculate the 2D Nd field (Section 3.2.1).**

We agree that this would be an ideal way forward, but in order to fully simulate the satellite it would be necessary to perform 3D radiative transfer upon the model fields in order to calculate reflectances at the relevant wavelengths and then to perform the retrievals of effective radius and optical depth on these (since these are used to obtain Nd). Unfortunately, this is a capability that we do not have available currently and which would also be very computationally costly. Observations in the region of our study (Painemal, 2011 & 2012) show that the satellite retrieval generally performs well against in-situ observations of Nd suggesting that a direct comparison to model Nd is justified. The satellite retrieval assumes that Nd is constant throughout the cloud depth and this assumption is well validated by the studies just mentioned. Such an assumption is consistent with the model fields for this cloud and so the choice of height used should not be very important. Therefore, we choose the height at which the liquid water content is the highest since this is likely most representative of a cloudy part of the profile. We also only consider columns with LWP>5 g m$^2$ in order to exclude non-cloudy columns. Given that there are reasonably large uncertainties in the observations for this quantity we feel that errors introduced by these methods are of secondary importance.

**6. Sections 3.2.3 and 3.2.4 should be simplified. Too much detail now.**

We have simplified Section 3.2.3 and especially 3.2.4 to remove a lot of the unnecessary detail. In section 3.2.3 we have removed some of the detail regarding the different POC regions. In Section 3.2.4 we have changed some of the details due to the new figure being implemented (see response 7) and have cut out the detail regarding the Poisson errors and some of the quantitative quotes, which we deemed a little too complicated and did not add much beyond what could be ascertained from the figure itself. Please see the revised manuscript for more details.

**7. Why don't you use all available satellite data to plot the LWP PDF in Fig. 8? The**

**data sample in current Fig. 8 is very limited based on just one snapshot.**

Thanks for the suggestion - we have now revised Fig. 8 to include 9 snapshots from the REMSS microwave satellite instruments for the nighttime (the time centred around the maxima of the LWP diurnal cycle) PDF (no GOES data available). For the daytime (surrounding the LWP minima) there are only 4 available REMSS snapshots, but we have also sampled the GOES LWP (available every 30 minutes) for a longer period and for both of the days available. Here is the revised figure and caption :-

[Figure]

**Figure 8.** PDFs of LWP for daytime (left) and nighttime (right) time periods for the model and for satellite observations. "REMSS" refers to the several available REMSS microwave instruments, each of which provides a snapshot LWP field. For the daytime, the times surrounding the minima in the LWP diurnal cycle (see Fig 4) are used (10-18 LST on both 12th and 13th November; overall 4 REMSS snapshots). For the nighttime, only the REMSS satellites are shown; times are chosen surrounding the maxima of the LWP cycle, but the surrounding period is reduced compared to the daytime in order to match the limited available REMSS times as closely as possible (03-09 LST on 12th Nov, 20 LST on 12th Nov to 10 LST on 13th Nov, 18:30 - 20 LST on 13th Nov; contains 9 REMSS snapshots). The model and GOES-10 data have been coarse grained to the AMSR-E resolution of 0.25°.

The description in Section 3.2.4 has also been adjusted accordingly (and simplified as requested in the previous comment).

**8.The 1X1 degree region in the model is too large to compare with the ship-borne radar**

**CFAD. The W-band radar will not cover such a big area. A cloud regime matching**

**technique can be used to choose the right area in the model results.**

We acknowledge that a 1x1 degree region is much larger than the size of the region sampled by the ship radar in a single profile. However, the radar is sampling continuously over time and thus will be capturing some spatial variability of the clouds as they move with the wind over the ship. The model data is available every 30 minutes (ship data every 0.3 seconds) and so given the limited time frequency of the model some spatial sampling is warranted (assuming that spatial sampling can make up for a lack of temporal sampling). Using a wind speed of 10 m/s the cloud field will move by 18km in 30 minutes and so our choice of 1x1 degree is likely to lead to a larger sampling region than was the case in reality. However, tests using smaller sampling regions (down to 3x3km; not shown) for the model show very little change in the patterns of frequencies for the CFADs indicating that this choice of sampling scale is not very important for this comparison.

We have added a comment on this in the paper at p21, line 1 :-

Tests using smaller sampling regions (down to $3 \times 3$ km; not shown) show very little change in the patterns of frequencies for the 2D PDFs indicating that the choice of sampling scale is not very important for this comparison.

**9. The RHcrit in the sub-grid cloud scheme should be a function of aerosol loading so that the aerosol impact on sub-grid cloud can be addressed.**

We are a little unclear about what is meant here. The RHcrit parameter and the cloud scheme is intended to simulate the extra condensation of vapour into cloud liquid due to sub-grid variability of relative humidity. The condensation of vapour into cloud does not depend on aerosol for the resolved scheme and so this is also the case for the sub-grid cloud scheme. The sub-grid cloud allows a response to aerosol via the microphysics, which takes into account any sub-grid cloud fraction, so that the in-cloud droplet concentrations (and liquid water contents) are enhanced relative to the grid-box mean, if the cloud fraction is less than one. This allows the aerosol to affect the sub-grid precipitation formation for example.

***Some technical suggestions:***

**Page 1, lines 18 to 19 and later in Page 24: ... to within delta fc = 0.04 ... It is the fc**

**difference not the fc itself.**

Fixed.

**Page 5, line 23: ... since the presence of some liquid ...**

Fixed.

**Page 7, line 2: ... investigate its impact. You shouldn't know whether it is important or**

**not.**

Fixed.

**Page 8, caption of Figure 2: should the box be blue not white?**

Fixed.

**Page 17, line 20 and page 19 last line: ... for the entire of the ...**

Fixed.

**Page 20, line 5: ... similar to those observed by CERES.**

Fixed.

**Page 20, line 6: ... unlikely to be the result of cloud fraction ...**

Fixed.

**Page 20, line 7: ... distribution for this and higher ...**

Fixed.

**Page 26, line 6: ... and that leads to lower boundary layer.**

Fixed.

**Page 31, line 16: Please reword this sentence.**

Done.

**Page 31, line 30: , which was responsible ...**

Fixed.